# CM1-driven assembly and activation of yeast γ-tubulin small complex underlies microtubule nucleation

Axel F Brilot[1], Andrew S Lyon[1†‡], Alex Zelter[2], Shruthi Viswanath[3§], Alison Maxwell[1], Michael J MacCoss[4], Eric G Muller[2], Andrej Sali[3], Trisha N Davis[2], David A Agard[1]*

[1]Department of Biochemistry and Biophysics, University of California at San Francisco, San Francisco, United States; [2]Department of Biochemistry, University of Washington, Seattle, United States; [3]Department of Bioengineering and Therapeutic Sciences, University of California at San Francisco, San Francisco, United States; [4]Department of Genome Sciences, University of Washington, Seattle, United States

**\*For correspondence:**
agard@msg.ucsf.edu

**Present address:** †Department of Biophysics and Howard Hughes Medical Institute, University of Texas Southwestern Medical Center, Dallas, United States; ‡Howard Hughes Medical Institute, University of Texas Southwestern Medical Center, Dallas, United States; § National Center for Biological Sciences, Tata Institute of Fundamental Research, Bangalore, India

**Competing interests:** The authors declare that no competing interests exist.

**Abstract** Microtubule (MT) nucleation is regulated by the γ-tubulin ring complex (γTuRC), conserved from yeast to humans. In *Saccharomyces cerevisiae*, γTuRC is composed of seven identical γ-tubulin small complex (γTuSC) sub-assemblies, which associate helically to template MT growth. γTuRC assembly provides a key point of regulation for the MT cytoskeleton. Here, we combine crosslinking mass spectrometry, X-ray crystallography, and cryo-EM structures of both monomeric and dimeric γTuSCs, and open and closed helical γTuRC assemblies in complex with Spc110p to elucidate the mechanisms of γTuRC assembly. γTuRC assembly is substantially aided by the evolutionarily conserved CM1 motif in Spc110p spanning a pair of adjacent γTuSCs. By providing the highest resolution and most complete views of any γTuSC assembly, our structures allow phosphorylation sites to be mapped, surprisingly suggesting that they are mostly inhibitory. A comparison of our structures with the CM1 binding site in the human γTuRC structure at the interface between GCP2 and GCP6 allows for the interpretation of significant structural changes arising from CM1 helix binding to metazoan γTuRC.

## Introduction

The microtubule (MT) cytoskeleton plays an essential role in the spatio-temporal control of eukaryotic cellular organization, cytoplasmic transport, and chromosome segregation during mitosis (*Desai and Mitchison, 1997*). The organization and function of the cytoskeletal network is tightly controlled by regulating the rate and location of nucleation, as well as MT polymerization kinetics and stability (*Akhmanova and Steinmetz, 2015*; *Howard and Hyman, 2009*; *Teixidó-Travesa et al., 2012*).

In most cells, MT nucleation occurs primarily at MT organizing centers such as centrosomes or spindle pole bodies and is dependent on the universally conserved γ-tubulin ring complex (γTuRC) (*Lüders and Stearns, 2007*). In budding yeast, homologues of the grip-containing proteins (GCPs) GCP2 and GCP3 (Spc97p and Spc98p) and two copies of γ–tubulin (Tub4p) form a 300 kDa complex (γTuSC) (*Vinh et al., 2002*). Spc110p, a distant pericentrin homologue, recruits this complex to the nuclear face of the spindle pole body (SPB), while Spc72p recruits it to the cytoplasmic face (*Knop and Schiebel, 1997*; *Knop and Schiebel, 1998*; *Nguyen et al., 1998*). Both of the γ-tubulin complex receptors contain the highly conserved centrosomin motif 1 (CM1) (*Lin et al., 2014*; *Zhang and Megraw, 2007*). In yeast, the CM1 motif is required for seven identical γTuSCs to

helically assemble into a γTuRC at the SPB (*Knop and Schiebel, 1998*; *Kollman et al., 2015*; *Lyon et al., 2016*; *Nguyen et al., 1998*).

In metazoans and plants, the γTuRC is recruited to MT nucleation sites as a large, pre-formed ring-shaped 2.2 MDa complex (*Teixidó-Travesa et al., 2012*). The metazoan γTuRC is composed of 14 γ-tubulins, and a smaller number of the γ-tubulin binding proteins, GCP2-6, as well as other accessory proteins. While sharing only ~15% homology and varying in size from 70 kDa to 210 kDa, GCP2-6 share a conserved core of two grip domains (*Guillet et al., 2011*). Structural and biochemical studies have shown that the N-terminal grip1 domain drives lateral association between GCPs, while the grip2 domain binds to γ-tubulin (*Choy et al., 2009*; *Farache et al., 2016*; *Greenberg et al., 2016*; *Guillet et al., 2011*; *Kollman et al., 2015*). Recent cryo-EM structures revealed that five copies of the GCP2/3 γTuSC are integrated into the metazoan γTuRC along with a GCP4/5 and a GCP4/6 pseudo-γTuSC as well as other accessory proteins (*Consolati et al., 2020*; *Liu et al., 2020*; *Murphy et al., 2001*; *Oegema et al., 1999*; *Wieczorek et al., 2020b*). In the human γTuRC, two copies of the CM1-containing CDK5RAP2 γTuRC nucleation activator (γTuNA) were found at the GCP2-6 interface (*Wieczorek et al., 2020a*).

Previous moderate-resolution cryo-EM structural studies (8 Å) had shown that wild-type (WT) yeast γTuSCs complexed with the N-terminal domain of Spc110p self-assemble into helical filaments (hereafter γTuRC) having 6.5 γTuSCs/turn, thereby presenting 13 γ-tubulins to template 13-protofilament MTs (*Kollman et al., 2010*; *Kollman et al., 2015*). Although close to MT symmetry, the γ-tubulins within each γTuSC were too far apart to correctly match the MT lattice, adopting an open conformation. The relevant in vivo conformation was determined by cryo-tomography and sub-volume averaging, clearly showing a MT-matching geometry at the yeast SPB, suggesting that γTuRC closure might be an important regulatory step (*Kollman et al., 2015*). To validate this hypothesis, γ-tubulin was engineered with disulfides to stabilize a closed MT-like conformation (γTuRC^SS), resulting in significantly enhanced MT nucleation (*Kollman et al., 2015*). This also had the benefit of improving the cryo-EM map (6.5 Å) such that an initial pseudoatomic model (*Greenberg et al., 2016*; *Kollman et al., 2015*) could be built based on the crystal structure of human γ-tubulin (*Aldaz et al., 2005*; *Rice et al., 2008*) and the distant and much smaller (75 kDa vs. 97 or 98 kDa) human GCP4 (*Guillet et al., 2011*; *Kollman et al., 2015*). These structures suggest a hierarchical model of γTuSC activation, with γTuSC assembling at the SPB in an Spc110p-dependent manner into an open slightly active conformation of the γTuRC prior to γTuRC closure.

Biochemical studies on the role of Spc110p in γTuRC assembly revealed that higher-order oligomerization of Spc110p and its binding to γTuSCs was required to overcome the intrinsically weak lateral association of γTuSCs at physiologically relevant γTuSC concentrations (*Kollman et al., 2010*; *Kollman et al., 2015*; *Lyon et al., 2016*). Deletion studies identified that independent of oligomerization removal of Spc110p residues 1–111 (Spc110p^1-111) was lethal in vivo but only slightly compromised γTuRC assembly in vitro, perhaps suggesting an essential regulatory function. By contrast, deletion of the subsequent centrosomin motif 1 (CM1, Spc110p^117-146) additionally abolished γTuRC assembly in vitro (*Figure 1A*). Supporting the need for precise regulation of γTuRC assembly and function, all the components of the γTuSC, as well as Spc110p and Spc72p, are phosphorylated at multiple sites in a cell-cycle-dependent manner (*Fong et al., 2018*; *Keck et al., 2011*). Mutations at several of these phosphorylation sites have been shown to impact cellular viability, spindle morphology, or shown to affect γTuRC assembly (*Fong et al., 2018*; *Huisman et al., 2007*; *Keck et al., 2011*; *Lin et al., 2011*; *Lin et al., 2014*; *Lyon et al., 2016*; *Peng et al., 2015*; *Vogel et al., 2001*). Together these data suggest a hierarchical model of γTuSC activation; with γTuSC assembling at the SPB into an open γTuRC that would be further activated by closure (*Figure 1B*). However, owing to the lack of structural data, the molecular mechanisms by which Spc110p facilitates γTuRC assembly and activation have remained unclear.

Here, we use crosslinking mass spectrometry (XL-MS) and X-ray crystallography to identify and determine the structure of the N-terminal coiled-coil (NCC) of Spc110p-bound in γTuSC filaments previously observed in cryo-EM reconstructions. The combined data show that a unique pose of the coiled-coil satisfies most of the XL-MS restraints. Furthermore, integrative modeling indicates that only residues N-terminal to the coiled-coil from a single protomer were required to satisfy the majority of the crosslink restraints, suggesting an asymmetric mode of Spc110p binding to γTuRC.

We present cryo-EM structures of monomeric and dimeric γTuSCs at near-atomic resolution, as well as higher-resolution (~3.0–4.0 Å) cryo-EM structures obtained from γTuRC filaments in the open

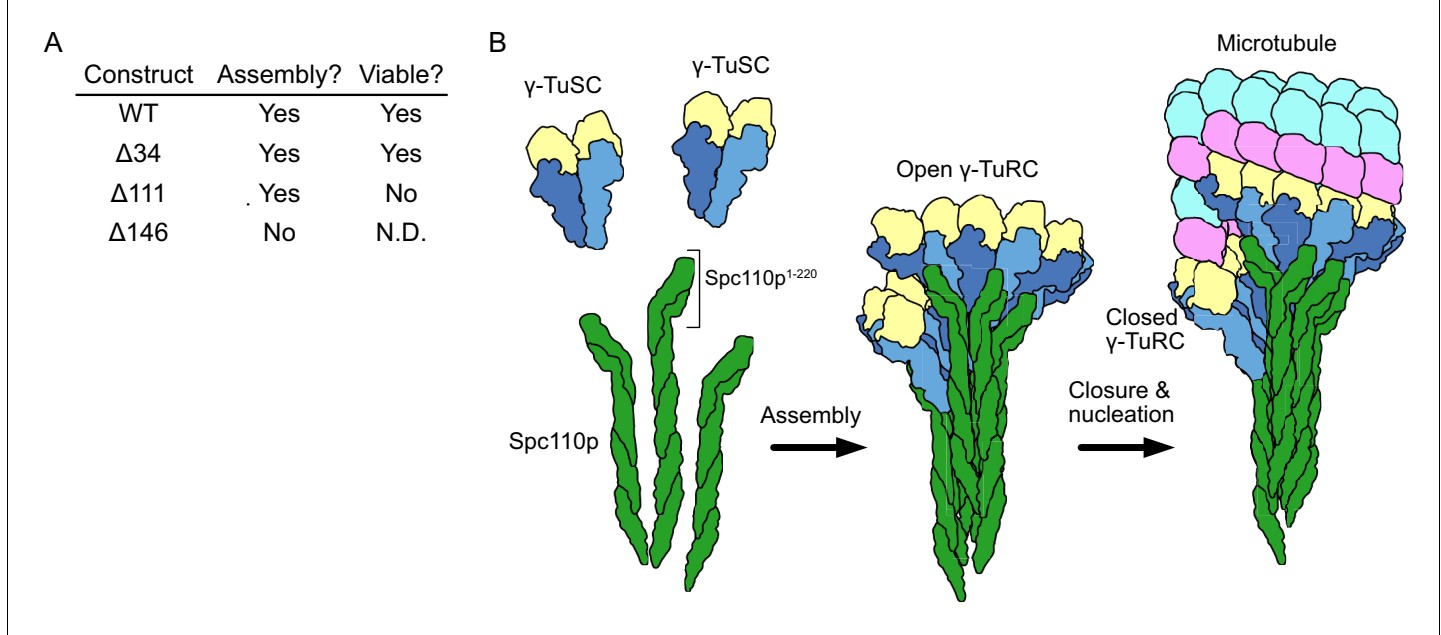

**Figure 1.** Spc110p and γ-tubulin small complexes (γTuSCs) assemble to form γ-tubulin ring complexes (γTuRCs) prior to microtubule nucleation. (**A**) An overview of the effect of Spc110p deletions on assembly and viability, summarizing previously published data from *Lyon et al., 2016*. Assembly data was generated using Spc110p-GCN4 tetramer fusion constructions, while in vivo data used full-length proteins in a red-white plasmid shuffle assay. (**B**) Schematic overview of γTuRC assembly: monomeric γTuSCs bind to Spc110p$^{1-220}$ and assemble into an open γTuRC, which undergoes closure prior to or concurrent with microtubule nucleation.

and closed conformations (*Kollman et al., 2010*; *Kollman et al., 2015*). These have allowed de novo model building of unknown regions and reinterpretation of significant portions of γTuSC structure. Our atomic models of γTuSCs in different assembly and conformational states provide insights into the mechanisms of γTuRC assembly and activation mechanisms required for MT nucleation and reveal how N-terminal regions of Spc110p, notably CM1, facilitate γTuRC assembly. Many of the annotated phosphorylation sites had fallen in regions of γTuSC not present in GCP4, and hence not previously modeled. Thus, the new structure provides a powerful atomic framework for understanding the importance and mechanism of regulatory modifications.

## Results

### Defining Spc110p:γTuSC interactions by XL-MS

Our previous 6.9 Å cryo-EM reconstruction, derived from helical filaments of Spc110p bound to an engineered closed conformation of γTuRC$^{SS}$, revealed an ~40-residue-long segment of coiled-coil density contacting the N-terminal region of Spc97p. The limited resolution prevented rigorous assignment of this density to any particular portion of Spc110p. Given its importance for γTuRC assembly, the coiled-coil seemed likely to correspond to either the conserved Spc110p$^{CM1}$ or the 45-residue segment (Spc110p$^{164-208}$) predicted with high probability to be a coiled-coil (the NCC, or Spc110p$^{NCC}$; see *Figure 2A*). Beyond this ambiguity, previous maps also lacked any density for the non-coiled-coil regions of Spc110p$^{1-220}$ known to be biochemically important and absolutely required for viability (*Lyon et al., 2016*).

To define the important interaction interfaces between Spc110p and γTuSC, we utilized XL-MS with the same Spc110$^{1-220}$ construct used for the cryo-EM as well as a longer Spc110$^{1-401}$ construct (*Figure 2—figure supplement 1*). We observed a significant number of crosslinks between the N-terminal portions of Spc97p and Spc110p$^{NCC}$ (*Figure 2—figure supplement 1*). Thus, this region and not CM1 was responsible for the coiled-coil-γTuSC interaction apparent in the cryo-EM map. As shown below, CM1 instead binds at a cleft that spans two adjacent γTuSCs.

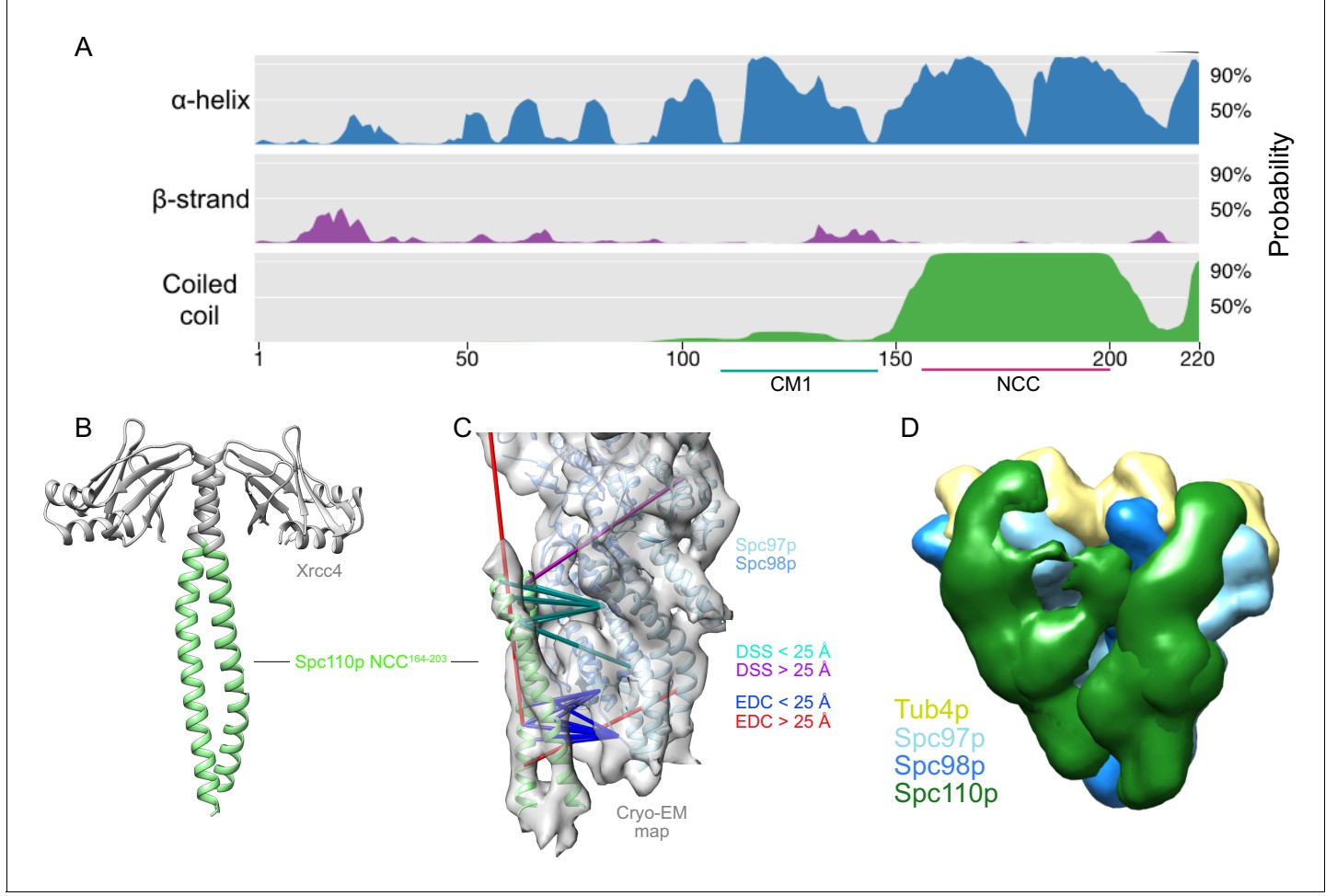

**Figure 2.** The Spc110p[NCC] binds near the N-terminus of Spc97p. (**A**) Spc110p N-terminal region secondary structure prediction, showing lack of predicted secondary structure for the first 111 residues. Also shown are Spc110p[CM1(117-146)] and the Spc110p[NCC(164-208)] regions. (**B**) Structure of Xrcc4-Spc110p[164-207], where Spc110p[NCC] residues 164–203 are resolved. (**C**) Spc110p[NCC] structure fit into γ-tubulin ring complex (γTuRC) cryo-EM density map (gray surface, EMDB ID 2799) along with γ-tubulin small complex (γTuSC) pseudo-atomic model (PDB ID 5FLZ) (*Kollman et al., 2015*; *Greenberg et al., 2016*). The majority of crosslinking mass spectrometry (XL-MS) distance restraints are satisfied by this model. Satisfied and violated disuccinimidyl suberate (DSS) crosslinks are shown in cyan and purple, respectively. Satisfied and violated 1-ethyl-3-(3-dimethylaminopropyl) carbodiimide (EDC) crosslinks are shown in blue and red, respectively. Crosslinks that are satisfied by either Spc110p monomer are shown twice, one for each monomer. (**D**) Localization density map for the ensemble of integrative models consisting of two adjacent γTuSCs, each bound to an Spc110p[1-220] dimer. The map shows the positions of different parts of the complex in the ensemble of models from the top cluster; maps for all components are contoured at 2.5% of their respective maximum voxel values. The modeling results shown are based on the γTuSC-Spc110p[1-220]-GCN4 crosslinks; similar results were obtained using γTuSC-Spc110p[1-401]-GST crosslinks (see Appendix 1).

The online version of this article includes the following figure supplement(s) for figure 2:

**Figure supplement 1.** Overview of crosslinking mass spectrometry (XL-MS) datasets.

**Figure supplement 2.** Results of integrative modeling of the Spc110p-γ-tubulin small complex (γTuSC) complex.

**Figure supplement 3.** The four stages of integrative modeling of the Spc110-γ-tubulin small complex (γTuSC) complex.

**Figure supplement 4.** Results for sampling exhaustiveness protocol for modeling the complex of Spc110p[1-220]-GCN4 dimer with γ-tubulin small complex (γTuSC).

## The Spc110p NCC[164-208] binds to γTuSC at the N-terminal regions of Spc97p

Due to the limited resolution of the previous cryo-EM reconstruction, the derived atomic model of the coiled-coil contained only the peptide backbone (*Greenberg et al., 2016*; *Kollman et al., 2015*). Motivated by the crosslinks observed between the Spc110p[NCC] and γTuSC, we sought a higher-resolution structure of the NCC region via X-ray crystallography. Previous work indicated that

Spc110$^{1-220}$ is only weakly dimeric (*Lyon et al., 2016*). Using the proven strategy of fusing weakly interacting coiled-coils with stabilizing domains (*Andreas et al., 2017*; *Frye et al., 2010*; *Klenchin et al., 2011*), we found that an N-terminal fusion of Spc110p$^{164-207}$ with a domain from Xrcc4 produced high yields of soluble protein. The Xrcc4-Spc110p$^{164-207}$ construct crystallized in a variety of conditions, diffracted to 2.1 Å, and enabled phases to be obtained by molecular replacement using Xrcc4 as a search model. As expected, the electron density map was consistent with a coiled-coil, with interpretable density for Spc110p$^{164-203}$ residues (*Figure 2B*, *Supplementary file 1*). When the coiled-coil was docked into the 6.9 Å cryo-EM map using cross-correlation in Chimera (*Pettersen et al., 2004*), the X-ray model occupied most of the alpha-helical cryo-EM density. Importantly in the docked conformation the majority of the unique crosslinks (*Figure 2C*, *Figure 2—figure supplement 1E*) were satisfied.

To better understand where the non-coil regions of Spc110p might interact, we used integrative modeling (*Figure 2D*, *Figure 2—figure supplement 2*, *Figure 2—figure supplement 3*, Appendix 1) (*Alber et al., 2007*; *Greenberg et al., 2016*; *Kollman et al., 2015*; *Rout and Sali, 2019*; *Russel et al., 2012*).

We first considered a single γTuSC bound to an Spc110p$^{1-220}$ dimer (*Figure 2—figure supplement 2A*). Using a combination of the previous cryo-EM-based γTuSC pseudoatomic model (*Greenberg et al., 2016*; *Kollman et al., 2015*), the X-ray structure of Spc110p$^{164-207}$, and representing the rest of γTuSC and Spc110p$^{1-220}$ by flexible strings of beads representing the amino acid chain, approximately 3000 good-scoring models were obtained satisfying the crosslinks and stereochemistry (excluded volume and sequence connectivity). These models were clustered based on structural similarity (*Figure 2—figure supplement 4*), and ~98% of the models were well represented by a single cluster that satisfied >90% of the crosslinks (see Appendix 1).

Consistent with visual inspection of the crosslinks, the localization probability density map from the most occupied cluster (*Figure 2—figure supplement 2A*) indicated that Spc110p$^{1-163}$ extended from the Spc110p$^{NCC}$ along the Spc97p-Spc98p interface towards γ-tubulin and the C-termini of Spc97p/98p. The precision of the model was insufficient to distinguish separate paths for the non-coiled-coil regions of each protomer within the Spc110p dimer. Consequently, we also considered a model containing Spc110p$^{1-163}$ from a single protomer, which almost equally satisfied the crosslink restraints and indicated a similar path (*Figure 2—figure supplement 2B*).

As the localization probability map suggested that the two Spc110p protomers might follow different paths, with one path extending towards the adjacent γTuSC, we also modeled two adjacent γTuSCs, each bound to an Spc110p$^{1-220}$ dimer (*Figure 2D*). By considering adjacent γTuSCs, the predicted path spans from the N-terminus of Spc97p of one γTuSC before proceeding towards the Spc98p from the adjacent γTuSC and binding in the space between the two γTuSCs. There is also a component that extends towards the Spc97p C-terminus and γ-tubulin (*Figure 2D*). Together these results suggest a complex path interacting with multiple γTuSCs taken by at least one of the two Spc110p N-termini.

## High-resolution filament structures reveal previously uninterpretable regions of γTuSC

The observed binding site between Spc110p$^{NCC}$ and γTuSC explains how Spc110p oligomerized at spindle poles can stimulate γTuRC assembly by increasing the local γTuSC concentration. However, this fails to explain the critical biochemical and in vivo functional importance of residues N-terminal to the Spc110p$^{NCC}$, such as the Spc110p$^{CM1}$ region, for γTuRC assembly and MT nucleation (*Lyon et al., 2016*). While the crosslinking and integrative modeling data suggested a physical basis for these functional roles, the actual path and interactions taken by Spc110p$^{1-163}$ were unknown. Realizing that this would require much higher resolution of Spc110p-γTuSC interactions, we focused on obtaining higher resolution structures of the 'open' (γTuRC$^{WT}$) and disulfide trapped 'closed' (γTuRC$^{SS}$) filaments containing Spc110p (*Kollman et al., 2010*; *Kollman et al., 2015*) by collecting new datasets on a direct electron detector and incorporating symmetry expansion and 3D classification into the data processing pipeline.

Filaments were initially processed in Relion2 (*Kimanius et al., 2016*), while allowing for the refinement of helical parameters, prior to further refinement of alignment parameters in FREALIGN (*Grigorieff, 2016*). As with previous studies, a combination of local helical and conformational inhomogeneities led to significantly worse resolution in the Spc97p/Spc98p C-terminus/γ-tubulin

region compared to the N-terminal and middle domains of Spc97p/Spc98p, particularly for the γTuRC$^{WT}$ filaments. To improve the resolution, we performed symmetry expansion followed by focused classification of segments containing three adjacent γTuSCs. The resulting reconstructions were at a resolution of 3.6 Å and 3.0 Å for the γTuRC$^{WT}$ and γTuRC$^{SS}$ filaments, respectively (*Figure 3A, B*, *Figure 3—figure supplements 1–3*, *Supplementary file 2*). The significantly increased resolution (*Figure 3—figure supplements 1–3*) allowed us to greatly improve upon previously published models of Spc97p, Spc98p, and γ-tubulin. Overall, we were able to build 712 a.a. of Spc97p (*Figure 3—figure supplement 4A*) (87%) 674 a.a. of Spc98p (*Figure 3—figure supplement 4B*) (80%), 453 a.a. of γ-tubulin (96%), and 95 a.a. of Spc110p$^{1-220}$ (43%).

Previous high-resolution crystal structures of γ-tubulin have shown that it adopts a bent-like state when not in complex with GCPs, independent of its nucleotide state (*Aldaz et al., 2005*; *Rice et al., 2008*). This raised the possibility that γ-tubulin might change conformation upon assembly into γTuSC. While the changes are small, in our structures, γ-tubulin adopts a conformation distinct from the previously observed bent human γ-tubulin or the yeast tubulin straight conformations (*Figure 3—figure supplement 5A*). In the assembled state, γ-tubulin H6 adopts what appears to be an intermediate conformation between the bent and straight conformations, while the C-terminal portion of the γ-tubulin$^{H6-H7}$ loop that most defines the interface with the incoming α-tubulin adopts a conformation similar to a straight yeast β-tubulin, likely potentiating MT formation (*Figure 3—figure supplement 5A*).

In looking for a potential cause for the altered γ-tubulin$^{H6-H7}$ loop conformation, there was one notable difference in the Spc/γ-tubulin interface. The γ-tubulin$^{T7}$ loop in assembled γ-tubulin moves such that it now more closely resembles the β-tubulin$^{T7}$ loop of an assembled β-tubulin (*Figure 3—figure supplement 5B*). The γ-tubulin$^{T7}$ loop is pinned between a loop (Spc98p$^{H15-16}$/Spc97p$^{H16-17}$) located at the N-terminus of a small domain in Spc97p and Spc98p and the adjacent C-terminal helical bundles (Spc98p$^{H22-23}$/Spc97p$^{H26-27}$). These results suggest that although subtle, assembly of yeast γ-tubulin into a γTuSC may help promote a more MT-like γ-tubulin plus end conformation, facilitating nucleation.

Previous structures of yeast GCPs and their assemblies suggested that the interface between the GCPs was largely formed from the two N-terminal helical bundles. Our high-resolution structures allow us to resolve large divergent N-terminal sequences present in both Spc97p and Spc98p, but absent in the shorter GCP4 'core' structure, which contribute to the intra- and inter-TuSC interfaces.

The GCP intra-γTuSC interface extends the entire length of the two N-terminal helical bundles of Spc97p and Spc98p, and also features significant contacts by the newly resolved N-terminal regions (*Figure 3C*). Of the residues newly modeled, Spc97p$^{1-54,81-89}$ and Spc98p$^{163-179}$ contribute an additional ~3600 Å$^2$ of buried surface area to the N-terminal interface. In addition, a previously unmodeled 33-residue insertion in the middle of Spc98p (Spc98p$^{672-704}$), between helices Spc98p$^{H23}$ and Spc98p$^{H24}$, folds into a pair of strands, contributing an additional ~1900 Å$^2$ of surface area. In the closed state, there is a small contact patch between the N-terminal region of Spc98p$^{H27}$ and Spc98p$^{H19}$. Thus, while the much shorter GCP4 structure, which formed the basis of previous modeling efforts, suggested well-conserved N-terminal interactions, it is clear that a very large part of intra-γTuSC stabilization (~5400 out of ~8000 Å$^2$, total interface) arises from sequences in Spc97p and Spc98p not present in GCP4, suggestive of very tight binding. This is consistent with γTuRC assembly, particularly smaller GCPs, being stabilized using non-γTuSC components in metazoans (*Liu et al., 2020*; *Wieczorek et al., 2020a*; *Wieczorek et al., 2020b*).

In contrast, the inter-γTuSC interface is much more limited in scope (total surface area ~2900 Å$^2$). It is mainly composed of two smaller, largely hydrophilic contact patches located at the three N-terminal helical bundles, and a small set of hydrophobic contacts. In addition, a small contact between Spc97p$^{K790}$ and Spc98p$^{Y510}$ (*Figure 3D*) involves almost no hydrophobic residues. The limited inter-γTuSC interface explains why γTuSCs fail to assemble under physiological concentrations (Kd ~2 μM), and thus must rely on a combination of CM1 interactions (see below) and avidity effects provided by Spc110p oligomerization (*Lyon et al., 2016*).

## Spc110p CM1 facilitates γTuRC assembly by binding at the inter-γTuSC interface

As before (*Figure 2B, C*), we observed coiled-coil density for the Spc110p$^{NCC}$ in our higher resolution maps. Given the observed pitch of the coiled-coil in the crystal structure, as well as density for

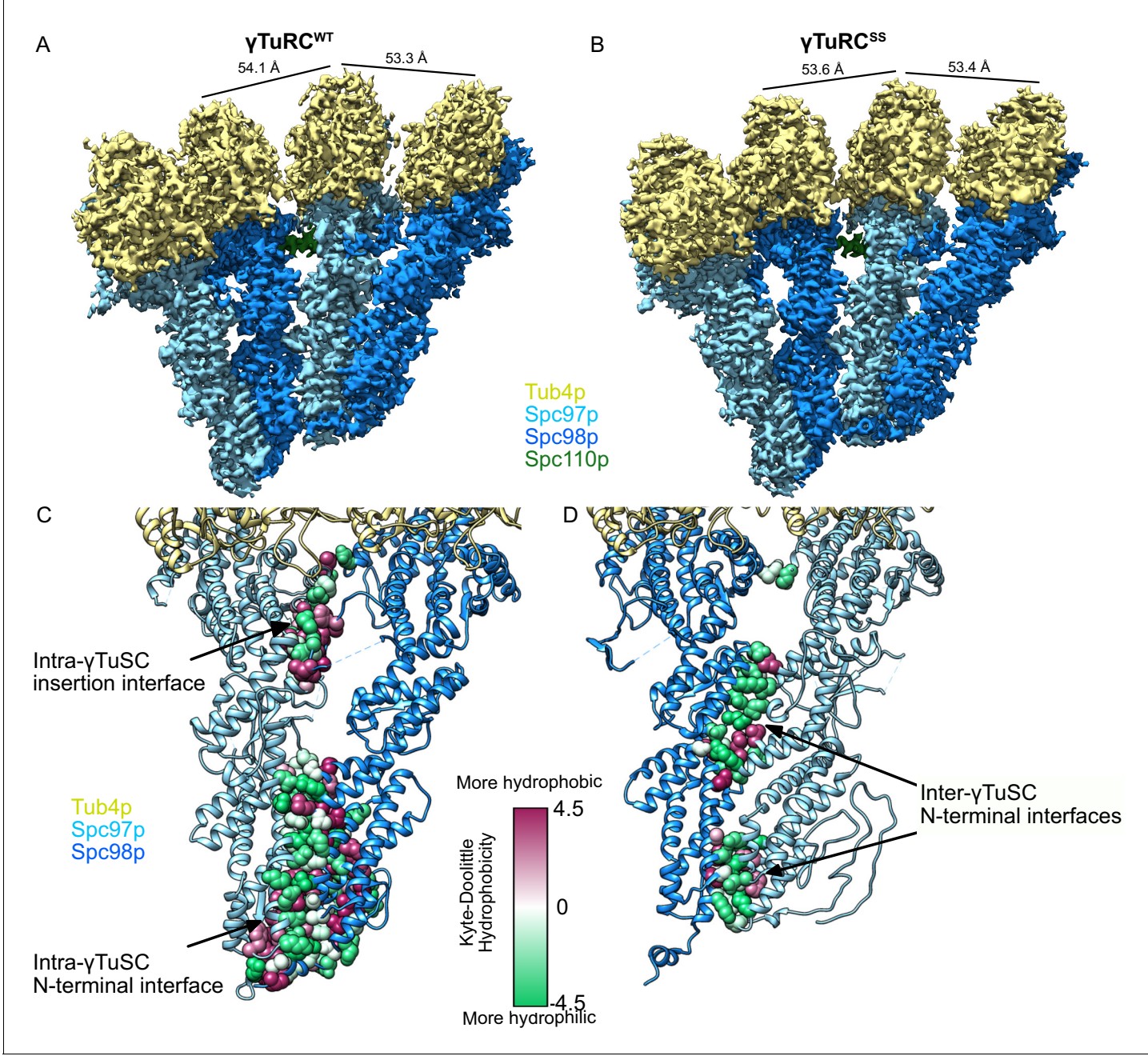

**Figure 3.** Structure and assembly interfaces of γTuRC^WT and γTuRC^SS. (A, B) Segmented density of (A) open γTuRC^WT and (B) closed γTuRC^SS. γTuRC subunits are colored as in the figure inset. Density was segmented within 4.5 Å of the atomic model, showing one Spc110p copy. Disconnected density smaller than 5 Å was hidden using the 'Hide Dust' command in Chimera. Spc110p^NCC is not visible at this threshold due to heterogeneity. (C, D) Representation of the intra- (C) and inter-γTuSC (D) interfaces of Spc97p/98p illustrated on a γTuRC^SS dimer. Interface atoms are shown as spheres and colored by their hydrophobicity according to the Kyte–Doolittle scale. γTuRC: γ-tubulin ring complex; γTuSC: γ-tubulin small complex.

The online version of this article includes the following figure supplement(s) for figure 3:

**Figure supplement 1.** γTuRC^WT processing and resolution.

**Figure supplement 2.** γTuRC^SS processing and resolution.

**Figure supplement 3.** γTuRC^SS and γTuRC^WT local resolution maps.

**Figure supplement 4.** Wiring diagrams of (A) Spc97p and (B) Spc98p.

**Figure supplement 5.** Comparison of γ-tubulin conformation between human and yeast γ-tubulin ring complex (γTuRC).

larger side chains, we were able to assign the register of the NCC (*Figure 4—figure supplement 1*). To assess the path of Spc110p N-terminal to the Spc110p$^{NCC}$, we generated a difference map between our experimental density maps and an atomic model for γTuRC, which did not include Spc110p atoms. This difference map should contain density for Spc110p and any regions not included in the atomic model. Indeed, the difference map revealed clear density extending from the NCC to a helical density that spans the inter-γTuSC interface and beyond (*Figure 4A*). Based on the side-chain features, we were able to unambiguously assign CM1$^{117-141}$ to the helical inter-γTuSC density (*Figure 4B*). While the density connecting the Spc110p$^{CM1}$ helix with Spc110p$^{NCC}$ was at a lower resolution, we were able to model residues Spc110p$^{112-206}$ spanning the Spc110p$^{CM1}$ and Spc110p$^{NCC}$ (*Figure 3—figure supplement 3C, D*).

Interestingly, a pair of helix-dipole/hydrogen bond interactions augment binding of the CM1 helix with Spc98p, with Spc98p$^{D542}$ hydrogen bonding with the N-terminus of the Spc110p$^{CM1}$ helix, and Spc110p$^{K120}$ hydrogen bonding with the C-terminus of helix Spc98p$^{H19}$ (*Figure 4—figure supplement 2A*). On Spc97p, the C-terminus of the Spc110p$^{CM1}$ helix interacts with helices Spc97p$^{H23}$ and Spc97p$^{H28}$ and the loop C-terminal to Spc97p$^{H21}$, as well as the insertion between Spc97p$^{H7}$ and Spc97p$^{H9}$ at the N-terminus of Spc97p$^{H8}$ (*Figure 4C*, *Figure 4—figure supplement 2B*). While we were unable to trace residues Spc110p$^{1-111}$ in our structure, numerous crosslinks map to the region between Spc97p and Spc98p and γ-tubulin at the intra-γTuSC interface (*Figure 2—figure supplement 2C*). These residues may therefore be involved in contacts facilitating activation and closure.

Together, our data reveal that one protomer of Spc110p$^{112-206}$ within each Spc110p$^{1-220}$ dimer adopts a complex path across two γTuSCs, while the Spc110p$^{112-165}$ region of the second protomer is unresolved, a path that defines the molecular role of the conserved CM1 motif. Beginning with Spc110$^{NCC}$ (Spc110p$^{164-208}$) bound to the N-terminus of Spc97p near the intra-γTuSC interface and moving towards the N-terminus, Spc110p next interacts with Spc98p and then weaves a path along the surface of Spc97p. From there, the CM1 helix binds across the inter-γTuSC interface to Spc98p on the adjacent γTuSC. After that, it continues along the surface of Spc98p, then turns towards the Spc97p C-terminus ending near γ-tubulin (*Figure 2—figure supplement 2D*, *Figure 4A, C*). Integrating these data generates a continuous path across two γTuSC subunits (*Figure 2—figure supplement 2D*). This is in good agreement with modeling predictions.

To assess the generality of the observed CM1 binding mode, we mapped conservation of the CM1 motif and its binding sites on Spc97p and Spc98p (*Figure 4—figure supplement 3A, B*). Of note, the more C-terminal portion of CM1 that binds to Spc97p is better conserved than the N-terminal portion that binds to Spc98p (*Figure 4—figure supplement 3C*). In keeping with this, the CM1 binding site on Spc97p is also highly conserved (*Figure 4—figure supplement 3B*). However, despite the limited conservation of the N-terminal portion of CM1, its binding site on Spc98p is well conserved in Spc98p/GCP3 homologues throughout eukaryotes (*Figure 4—figure supplement 3B*), attesting to its importance. Close inspection of the structure provides a molecular explanation: many of the interactions in this region are via CM1 backbone contacts and are thus less dependent on the precise CM1 sequence (*Figure 4—figure supplement 2*).

## Conformational changes of Spc97p and Spc98p during assembly

To better resolve fundamental questions about the molecular basis for γTuRC assembly and activation, we determined the cryo-EM structure of unassembled γTuSC, without Spc110p or filament formation, from images of frozen-hydrated single particles. At the concentration of ~1 μM used in data collection, micrographs and 2D classes show a mixture of γTuSC monomers and dimers, with a small number of larger oligomers (*Figure 5—figure supplement 1*). We were able to obtain a structure of the γTuSC monomer at ~3.7 Å, and of a γTuSC dimer at ~4.5 Å resolution (*Figure 5—figure supplements 2* and *3*, *Supplementary file 2*). The γTuSC dimer is formed from two γTuSCs in lateral contact using the same interface as observed in the γTuRC:Spc110p filament structures, but as expected lacks density for both the Spc110p$^{NCC}$ and the Spc110p$^{CM1}$ helix.

In order to assess the changes that occur during assembly of monomeric γTuSCs into the γTuRC and the subsequent closure, we aligned the N-terminal two helical bundles of Spc97p and Spc98p (residues Spc97p$^{52-276}$ and Spc98p$^{178-342}$). This alignment allows for a concise description of the joint conformational changes in both proteins that occur as γTuSCs assemble into γTuRCs and the subsequent closure required for MT nucleation (*Figure 5—figure supplement 4*, *Figure 5—videos 1–3*).

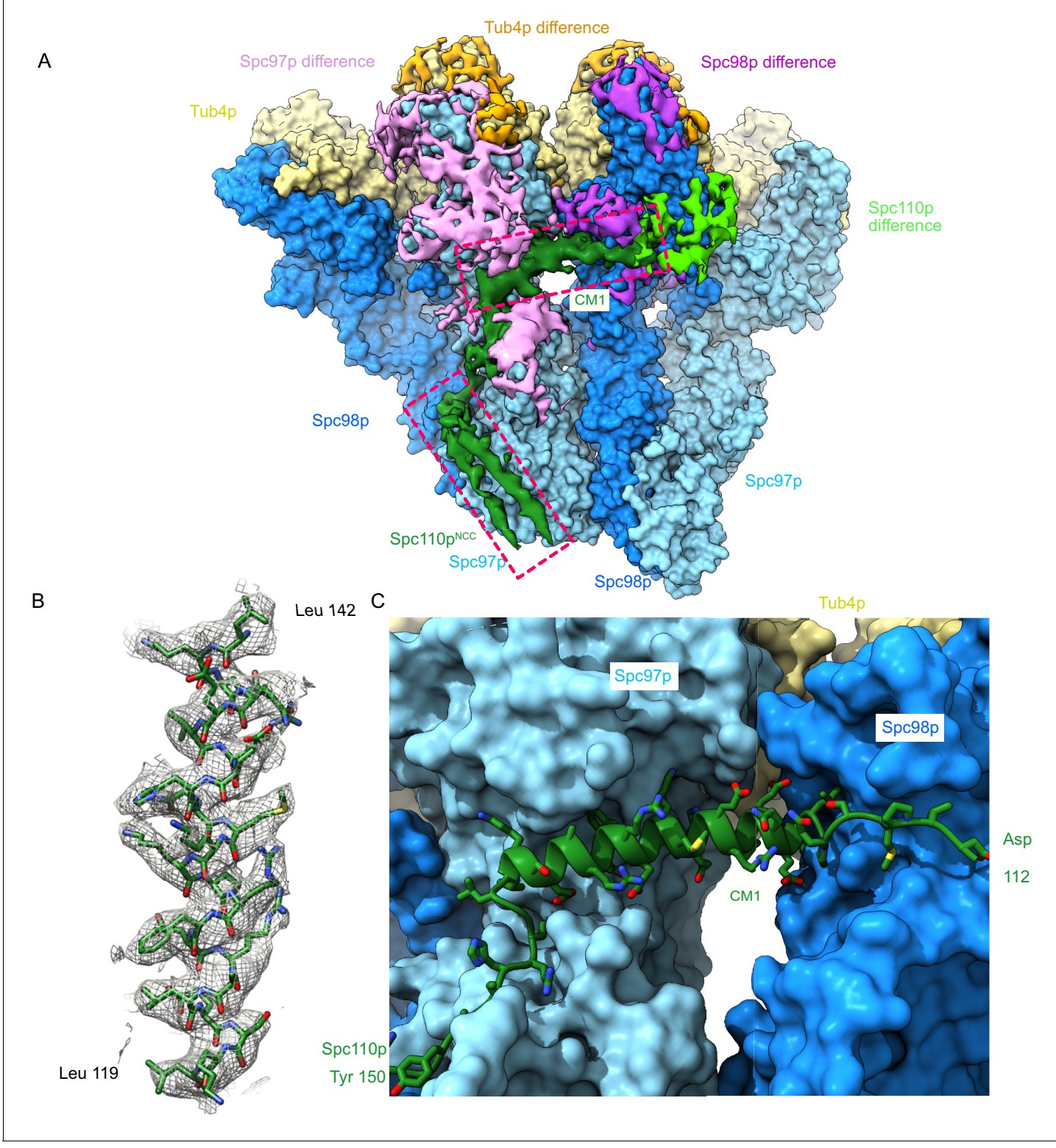

**Figure 4.** The Spc110p centrosomin motif 1 (CM1) helix binds at the inter-γ-tubulin small complex (γTuSC) interface. (**A**) Filtered segmented difference map between experimental density and the fitted atomic model without Spc110p overlaid on a γTuRC$^{SS}$ surface lacking Spc110p. The difference map was segmented to show density near a γTuRC$^{SS}$ monomer and colored to attribute densities to their putative chains. The Spc110p$^{NCC}$ and Spc110p$^{CM1}$ densities are highlighted with rectangular boxes. (**B**) Density for the helical CM1 density of γTuRC$^{SS}$ showing clear side-chain features unambiguously

*Figure 4 continued on next page*

Figure 4 continued

defining the register. Density was zoned near the atoms in Chimera with a radius of 2.6 Å. (C) View of the binding site for CM1 and the strands preceding and following the CM1 helix. γTuRC: γ-tubulin ring complex.

The online version of this article includes the following figure supplement(s) for figure 4:

**Figure supplement 1.** Spc110p$^{NCC}$ structure (forest green) near the tyrosine 186 side chain fitted into ~4.2 Å low pass filtered density from the γTuRC$^{SS}$ reconstruction. γTuRC: γ-tubulin ring complex.

**Figure supplement 2.** Helix dipole interactions define the centrosomin motif 1 (CM1) binding site on Spc98p.

**Figure supplement 3.** Conserved binding interface with the centrosomin motif 1 (CM1) motif.

During the transition from monomer to assembled open state (as seen in γTuRC$^{WT}$), the γ-tubulins move in the same overall direction, approximately orthogonal to the plane of the Spc97p/Spc98p contact interface (*Figure 5—figure supplement 4A*). The center of mass of the γ-tubulins shifts ~13.8 Å and ~15.6 Å when bound to Spc97p and Spc98p, respectively, as a result of twisting the helical bundles in Spc97p and Spc98p. All of the conserved contacts in Spc97p and Spc98p observed in assembled γTuSC filaments occur in the N-terminal three helical bundles. Notably, much of the bottom three helical bundles show only minor changes when assembling to the open state. The dominant changes occur on loop Spc98p$^{H10-S1}$ in the middle contact, which moves ~4.1 Å, and at the N-terminus of helix Spc98p$^{H11}$, involved in the top contact, which moves ~6.6 Å. The large conformational change in Spc98p that occurs to create these contacts in the γTuSC dimer is a major contributor to the even larger-scale changes during γTuRC assembly and activation.

## The transition from the open γTuRC$^{WT}$ to the closed γTuRC$^{SS}$

During the transition from the assembled open γTuRC$^{WT}$ to the engineered γTuRC$^{SS}$ closed conformation (*Figure 5*, *Figure 5—figure supplement 4B*), the γ-tubulins on Spc97p and Spc98p slide past each other in roughly opposite directions, undergoing translations of ~6.9 Å and ~7.7 Å, respectively (*Figure 5—figure supplement 4B*). In addition, the Spc98p-bound γ-tubulin undergoes a twisting motion of ~5–6°. During these conformational changes, the inter-γTuSC contacts make only minor alterations, mainly in the N-terminal three helical bundles of Spc97p and Spc98p, which undergo complex tilting and twisting motions. Overall, these conformational changes alter the pitch and twist of the γTuRC assemblies from ~140 Å/turn and 54.5° in the open state to ~132 Å/turn and 55.1° in the closed state (*Figure 5A, B, D, E*).

Excising a full turn in our γTuRC filament structures containing seven γTuSC subunits provides a good model for an isolated γTuRC as it might bind at the SPB. This reveals that within each γTuRC there are only six complete CM1 binding sites, the last one being interrupted at the end of the ring. This in turn suggests that only six Spc110p molecules need to be bound to a γTuRC in vivo to stabilize the full ring. This helps explain the apparent symmetry mismatch between the underlying hexameric organization of Spc42p (*Bullitt et al., 1997*; *Drennan et al., 2019*; *Muller et al., 2005*) within the SPB and the heptameric γTuRC. The geometry is such that the Spc110p$^{NCC}$ binding site most proximal to the SPB would be empty (*Figure 5E, F*).

Finally, by local 3D classification we observed that a closed state is populated in our γTuRC$^{WT}$ data (*Figure 5—figure supplement 1*, *Supplementary file 2*). This state had previously not been observed in the γTuRC$^{WT}$ structure as robust symmetry expansion and 3D classification techniques had not yet been developed for cryo-EM when the structure was published. While not identical to the disulfide crosslinked closed state, the differences are minimal, indicating that the conformational changes observed at highest resolution in γTuRC$^{SS}$ are representative of those occurring in the closed WT γTuRCs (*Figure 5—figure supplement 4C*). The fact that the WT closed state can occur spontaneously and is sampled in our open population suggests that, in the presence of Spc110p, the energy differences between the open and closed states are not large.

## Mapping phosphorylation sites on the γTuRC suggests largely inhibitory roles

The γTuSC is heavily phosphorylated in a cell-cycle-dependent manner, and perturbing phosphorylation has been shown to affect spindle morphology (*Fong et al., 2018*; *Keck et al., 2011*; *Lin et al., 2011*; *Peng et al., 2015*; *Vogel et al., 2001*). The role of many of these phosphorylation sites

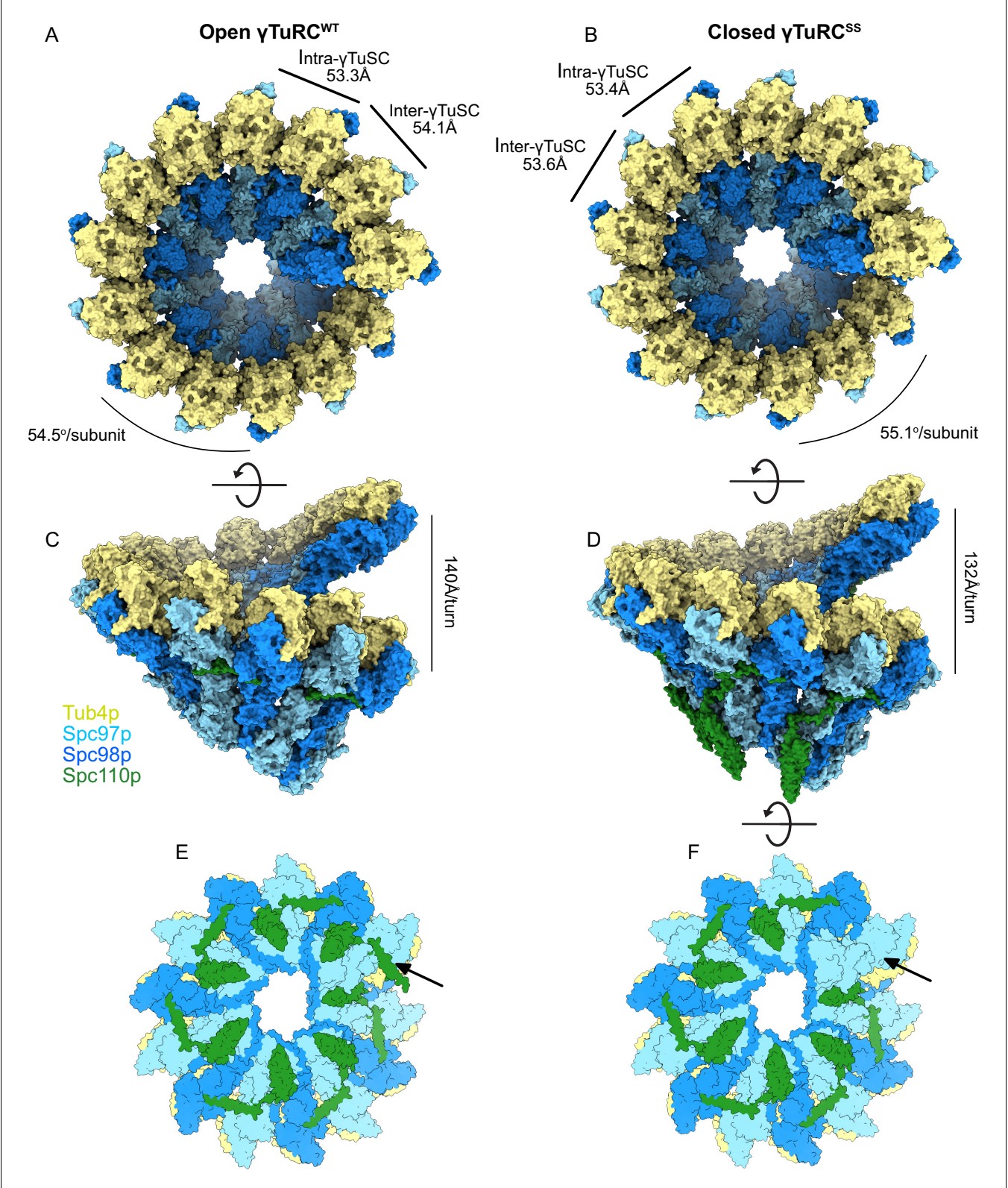

**Figure 5.** Structural overview of γ-tubulin ring complex (γTuRC) assemblies. Top and side views of open γTuRC^WT (A, C) and closed γTuRC^SS (B, D). Panels (E) and (F) show a bottom view of an assembled γTuRC^SS. The arrow indicates the seventh Spc110p^NCC binding site in the γ-tubulin small complex (γTuSC) heptamer, which is likely not to have bound Spc110p, given the sixfold symmetry observed in Spc42p at the spindle pole body (SPB),

*Figure 5 continued on next page*

*Figure 5 continued*

and the lack of a centrosomin motif 1 (CM1) binding site at the adjacent inter-γTuSC interface. Panel (E) shows the heptamer with a seventh Spc110p binding site, with CM1 only partially bound.

The online version of this article includes the following video and figure supplement(s) for figure 5:

**Figure supplement 1.** A mixture of compositional states is observed.

**Figure supplement 2.** WT γ-tubulin small complex (γTuSC) processing and resolution.

**Figure supplement 3.** Segmented single-particle reconstructions of γ-tubulin small complex (γTuSC) monomer and dimer.

**Figure supplement 4.** Conformational changes in γ-tubulin small complex (γTuSC) during assembly and activation.

**Figure 5—video 1.** Conformational changes of γ-tubulin small complex (γTuSC) during assembly.

https://elifesciences.org/articles/65168#fig5video1

**Figure 5—video 2.** Conformational changes of γ-tubulin ring complex (γTuRC) during activation.

https://elifesciences.org/articles/65168#fig5video2

**Figure 5—video 3.** Morph of γTuRC$^{SS}$ and γTuRC$^{WT}$ closed states shows minimal changes.

https://elifesciences.org/articles/65168#fig5video3

remains unclear as the phosphomimetic mutants used to investigate their function may not perfectly recapitulate the in vivo regulatory effects of the post-translational modifications. To better understand the potential role of phosphorylation in γTuRC assembly, regulation. and function, we mapped a recently determined set of phosphorylation sites, including a re-analysis of previously determined data and newly acquired data from SPBs (*Fong et al., 2018*), onto a dimer of our γTuRC$^{SS}$ structure (*Figure 6A*). Here, we focus on the serine/threonine sites, given the minimal tyrosine kinase activity in yeast. Surprisingly, phosphorylation at the majority of the mapped sites would seem to destabilize the assembled γTuRC and thus may help keep unassembled or partially assembled components inactive. Phosphorylation at two sites would likely stabilize assembly, indicating the complex modulatory role played by phosphorylation.

Many of these phosphorylation sites map to potentially important interfaces: the Spc110p/Spc97p interface (at the Spc110p$^{NCC}$ and at the Spc110p$^{NCC-CM1}$ loop), the inter-γTuSC interface, the γ-tubulin/α-tubulin interface, as well as a cluster of sites at the Spc97/98p:γ-tubulin interface. There are also a large number of unmapped phosphorylation sites, the majority of which are located on low-resolution or unresolved regions in the N-termini of Spc98p and Spc110p.

Strikingly, our γTuRC$^{SS}$ structure reveals a cluster of phosphorylation sites, with many exhibiting in vivo phenotypes, that maps near the Spc110p:γTuSC interface in the Spc110p$^{NCC}$ region and near the loop connecting Spc110p$^{NCC}$ and Spc110p$^{CM1}$. Of particular note are a set of sites on Spc110p (Spc110p$^{T182}$, Spc110p$^{T188}$) and the adjacent interface on Spc97p (Spc97p$^{S84}$, Spc97p$^{T88}$). Together, these would add numerous negative charges in a portion of the Spc97p/Spc110p interface that is already highly negatively charged, especially the Spc110p$^{NCC}$. Phosphorylation at two of these sites (Spc110p$^{T182}$ and Spc97p$^{S84}$) would likely negatively impact Spc110p binding, whereas Spc97p$^{T88}$ is adjacent to a positively charged patch; phosphorylation at this site would likely promote Spc110p binding.

Three sites on Spc97p (Spc97p$^{S130}$, Spc97p$^{S208}$, Spc97p$^{S209}$) and two sites on Spc110p (Spc110p$^{S153}$, Spc110p$^{S156}$) map onto or near the loop connecting Spc110p$^{CM1}$ with the Spc110p$^{NCC}$ and its interface with Spc97p. Mutation of Spc97p$^{S130}$ exhibited a temperature-sensitive phenotype, and the Spc97p$^{S208A/S209A}$, Spc97p$^{S208D/S209D}$ double mutants were lethal, consistent with phosphorylation of this region potentially having a regulatory role (*Fong et al., 2018*; *Lin et al., 2011*). While the loop has a lower resolution than other portions of the map (*Figure 3—figure supplement 3C, D*), the backbone approximately tracks with a long negatively charged patch along Spc97p and Spc98p (*Figure 6C*). Furthermore, the Spc110p$^{150-161}$ loop has two negative charges and one positive charge. Although phosphorylation at Spc110p$^{S153}$ and Spc110p$^{S156}$ was not consistently observed (*Fong et al., 2018*), phosphorylation at these sites, as well as on the opposite Spc97p interface, would likely destabilize Spc110p binding.

One site on Spc97p maps near the inter-γTuSC dimer interface. The interface is rearranged only by a few Ångstroms during activation, so any effect of phosphorylation would presumably only impact assembly. Spc97p$^{S797}$ mutations produce a mild phenotype (*Fong et al., 2018*), and it is unresolved in all of our structures, but it is likely on a flexible loop near a positive patch in Spc110p and Spc97p and may thus favor assembly.

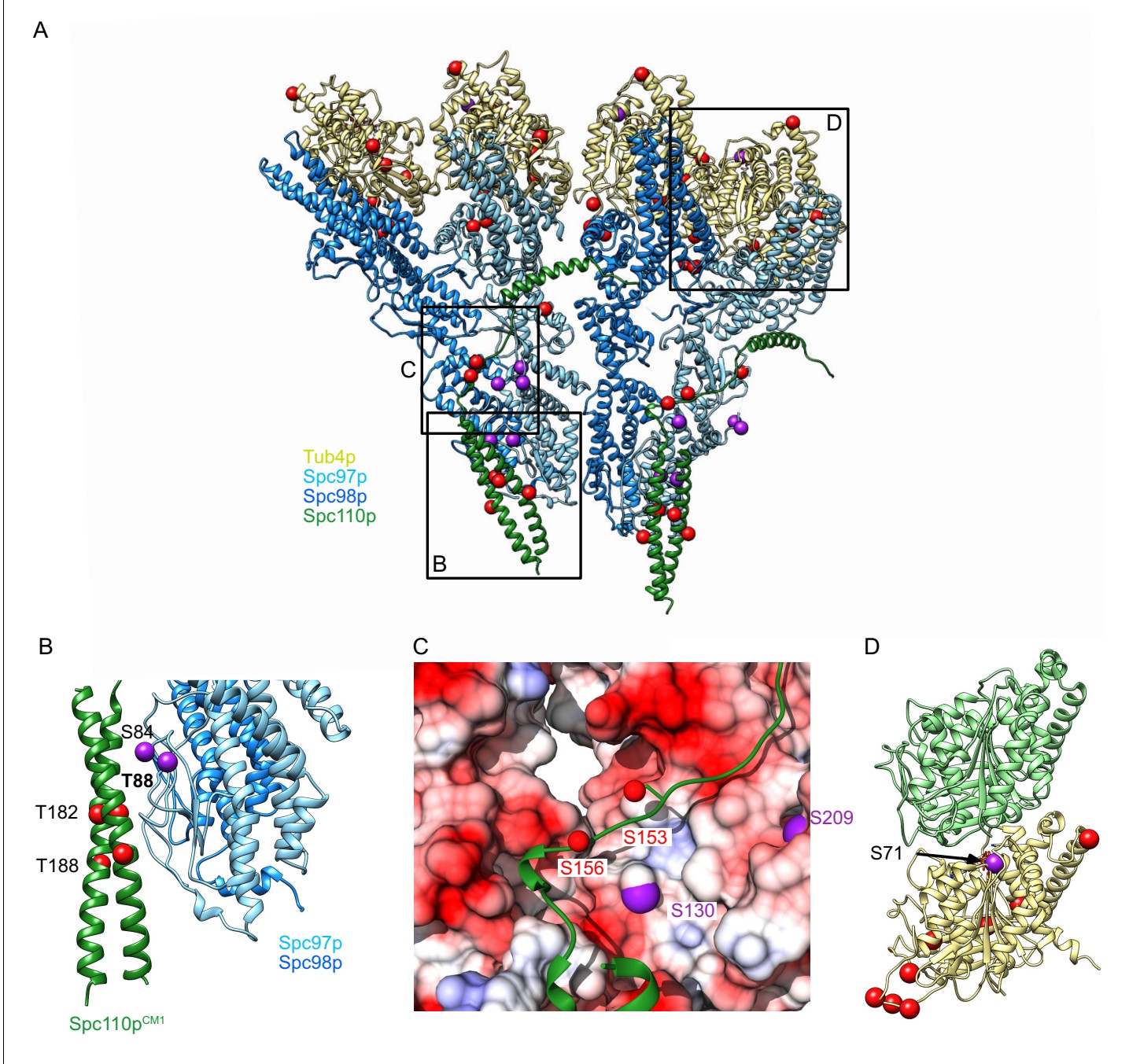

**Figure 6.** Phosphorylation sites visualized on the γTuRC[SS] structure. (**A**) γTuSC[SS] dimer, colored as in *Figure 2*, with phosphorylation sites from *Fong et al., 2018* marked with red balls (no known phenotype) or purple balls (phenotype previously reported). Boxes are shown highlighting areas shown in panels (**B–D**). (**B**) View of phosphorylation sites at the Spc97p Spc110p[NCC] binding site. The phosphorylation site T88 is labeled in bold as the only phosphorylation site localized at high resolution, which is expected to stabilize the interaction between Spc110p and the γ-tubulin ring complex (γTuRC) based on its proximity to a positive charge. (**C**) View of the path of the Spc110p loop between the Spc110p[NCC] and Spc110p[CM1] domain. This loop shows two phosphorylation sites opposite an acidic path. (**D**) Phosphorylation sites mapped on the γ-tubulin:α-tubulin interface, illustrating the position of the phosphorylation sites in relation to the interface with α-tubulin, Spc98p-bound γ-tubulin is in khaki, while α-tubulin is in light green.

Finally, γ-tubulin[S71] localizes near the γ-tubulin:α-tubulin interface, likely decreasing binding affinity, and perhaps even interfering with GTP binding. γ-tubulin[S71] and γ-tubulin[S74] mutants (A or D) both exhibit phenotypes, likely reflecting the importance of proper hydrogen bonding near the γ-tubulin GTP binding site (*Figure 6D*).

## Comparison of yeast γTuRC with metazoan TuRC structures

Recent efforts by several labs have been successful in providing the first models for the more complex metazoan γTuRCs (*Consolati et al., 2020*; *Liu et al., 2020*; *Wieczorek et al., 2020a*; *Wieczorek et al., 2020b*). These new structures provide much needed clarity on the stoichiometry of the five different GCPs (GCP2-6) and how they are organized within the γTuRC ring. They also reveal unexpected structural roles for numerous accessory components. Of interest to us was the role CM1-containing accessory proteins might have in metazoan γTuRC assembly and conformation.

Further support for a conserved role for CM1 is apparent in the recently published structure of the human γTuRC purified by affinity with γTuNA, an N-terminal truncation of CDK5RAP2, which includes its CM1 motif. The authors assign CM1 to the helical density at the interface between GCP2 and GCP6 (*Wieczorek et al., 2020a*; *Wieczorek et al., 2020b*). This is precisely equivalent to our assigned yeast CM1 helix at the Spc98p-Spc97p interface (*Figure 7—figure supplement 1A*). Notably, separate structural studies of human and *Xenopus* γTuRC, where the γTuRC was purified by affinity against GCP2 and γ-tubulin, respectively, showed no density at the same interface (*Consolati et al., 2020*; *Liu et al., 2020*). Furthermore, when the human γTuNA-bound map is filtered to low resolution, density similar to that observed in our yeast γTuRC[ss] difference map continues from the N-terminus of the CM1 helix along the surface of GCP6 towards GCP4 (*Figure 7—figure supplement 1B, C*). Taken together, these results stress the broad conservation and importance of CM1 binding.

These compositional differences led us to wonder whether CM1 binding might also drive conformational rearrangements in the metazoan γTuRCs, analogous to the changes we observed during yeast γTuRC assembly. Due to the lower resolution and lack of deposited atomic coordinates for the GCP2 affinity-purified human structures, we limited our comparison to the γTuNA-bound human and *Xenopus* (γTuNA-unbound) structures. Perhaps surprisingly, both metazoan γTuRC structures show a very poor match to MT symmetry and would require substantial γ-tubulin motions to match the MT (*Figure 7*). The γ-tubulins in the metazoan γTuRC structures are displaced up to ~46 Å from their ideal MT-like positions, as opposed to the 9 Å observed in our closed yeast γTuRC[ss] structure, suggesting that the metazoan γTuRCs may be even more strongly dependent upon additional factors or post-translational modifications (PTMs) to achieve an active conformation than the yeast γTuRCs.

While the human and *Xenopus* GCPs overlay very well at γ-tubulin positions 1–10, the terminal four positions show a different twist and pitch. We suggest here that these differences arise from CM1 binding at GCP2:GCP6 interface in the human γTuRC. The relative position of the γ-tubulins bound to GCP2 and GCP6 changes upon CM1 binding to much more closely match what we observe. That is, during the 'transition' from a CM1-absent γTuRC (*Xenopus*) to a CM1-present (human) γTuRC, the GCP6-bound γ-tubulin moves by ~10 Å to better match the position observed in our closed TuRC[ss] structure (*Figure 7—figure supplement 2A, B*). From this we speculate that binding of CM1-containing accessory proteins at other sites within the γTuRC would further optimize their conformation and MT nucleating ability. Within the centrosome, CM1-containing proteins are expected to be in very high local concentrations due to the highly colligative/phase condensation behavior of the pericentriolar material (*Feng et al., 2017*; *Woodruff et al., 2015*), further promoting γTuRC activation.

In contrast to the yeast structures, having CM1 bound in the human γTuRC seems to correlate with breaking the GCP2/6 N-terminal interface (*Figure 7—figure supplement 2C, D*). The dissociation of this N-terminal interface may be due to an intrinsically weaker GCP2/6 interaction, enhancing the role that additional factors that bind at CM1 or the inter-GCP interface could play in regulating MT nucleation.

## Discussion

Using a combination of single-particle and filament cryo-EM data, we have determined structures for monomeric and dimeric γTuSCs, along with assembled open and closed state γTuRCs at near-atomic

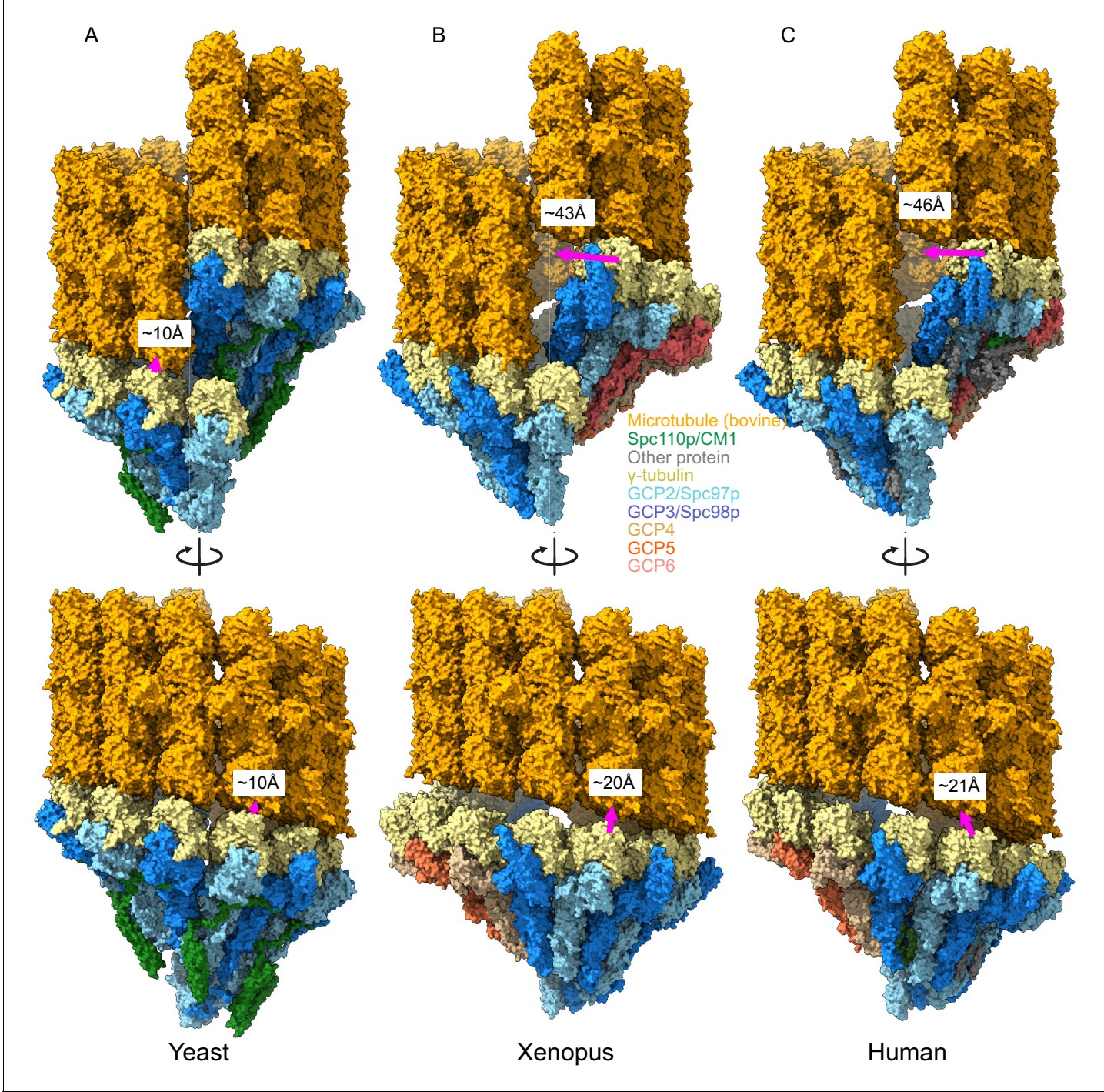

**Figure 7.** Metazoan γ-tubulin ring complexes (γTuRCs) require large motions to template microtubules. (**A**) Yeast closed (this work), (**B**) *Xenopus* (PDB ID 6TF9), and (**C**) human γTuRC (PDB ID 6V6S) structures placed adjacent to a microtubule to illustrate the motions required to properly template microtubules. For each structure, two γ-tubulins (positions 2, 3 for *Xenopus* and human and positions 13, 14 for yeast) were aligned with two β-tubulins docked in microtubule density to approximate binding of γ-tubulins to a microtubule.

The online version of this article includes the following figure supplement(s) for figure 7:

**Figure supplement 1.** A centrosomin motif 1 (CM1) helix binds between grip-containing protein (GCP)2 and GCP6 in human γ-tubulin ring complex (γTuRC).

**Figure supplement 2.** γ-Tubulin ring complex (γTuRC) undergoes large structural changes on centrosomin motif 1 (CM1) binding.

resolutions. Our structures complement existing structural and biochemical data with high-resolution snapshots of the yeast γTuSC and γTuRC. Together with previous work, these provide a framework for understanding the molecular basis for MT nucleation and regulatory processes likely necessary to ensure that MTs are only nucleated at the SPB. We provide the first molecular understanding for the critical role of the conserved Spc110p$^{CM1}$ region in yeast γTuRC assembly.

The structures suggest that nucleation is positively controlled in at least three ways: (1) assembly of γTuSCs into an open ring mediated by Spc110p oligomers and Spc110p$^{CM1}$, (2) closure of each γTuSC from an open state to a closed state to fully align the γ-tubulins to the MT lattice, and (3) phosphorylation at Spc110p$^{T88}$ can support Spc110p binding and directly impact γTuRC assembly. In addition, a number of phosphorylation sites on Spc110p and γ-tubulin would have a negative impact on assembly or MT nucleation, either inhibiting γTuSC binding to Spc110p or αβ-tubulin binding (*Figure 6*).

Although minor, we also observe conformational changes in γ-tubulin upon assembly into γTuSCs that mimic aspects of the bent-to-straight transition in αβ-tubulin and would thus be expected to enhance MT nucleation. Unresolved is to what extent these differences arise from differences in the protein sequence from yeast to metazoans or represent an assembly-driven enhancement in γ-tubulin conformation. There is at least some role for sequence as we know that there is a strong species barrier such that yeast γTuRC is hundreds-fold more potent at stimulating yeast tubulin polymerization than mammalian tubulin (*Kollman et al., 2015*).

As initially observed in negative stain EM (*Choy et al., 2009*), our new cryo-EM structures of monomeric and dimeric γTuSCs show that Spc97p and Spc98p intrinsically adopt an open conformation at the intra-γTuSC interface such that the attached γ-tubulins fail to make MT-like lateral contacts. Our structures of open and closed assembled γTuRCs show that Spc97p and Spc98p undergo large conformational changes during assembly into rings, bringing them much closer to MT geometry. Only smaller conformational changes occur as they transition from the open to the closed state during activation. The observation that a population of γTuRCs in the WT filaments adopts a locally closed conformation indicates a small energetic barrier to a single γTuSC closing, although simultaneous closing of an entire ring is unlikely. Thus, the addition/removal of PTMs or the binding of other factors could allosterically drive a more ideal template state. Indeed, we know that the yeast CK1δ kinase, Hrr25, is needed for proper spindle formation in vivo and that it binds to γTuRCs and stimulates MT assembly in vitro (*Peng et al., 2015*), indicating that it is one such activator. γTuRC closure may also be stabilized by the process of MT assembly.

Our structures also resolve a long-standing mystery: how the sixfold symmetric Spc42p layer at the SPB (*Bullitt et al., 1997*) could facilitate the formation of a γTuRC containing specifically seven γTuSCs (*Figure 8*). This is resolved by recognizing that Spc110p$^{CM1}$ within each dimer extends from one γTuSC to another, contributing to cooperative assembly, and cannot bridge across the large gap between the last and first γTuSCs in the ring. Thus, we suggest that six Spc110p dimers are symmetrically bound to the Spc42p lattice at the SPB. These would thus present the six CM1 motifs required to bind at the six complete CM1 binding sites formed within γTuRC heptamer. Given the observed pattern of connectivity where Spc110p CM1 extends across the interface in the same direction as the helical rise (*Figure 4A*), this would leave the terminal NCC site nearest to the SPB unoccupied.

Our high-resolution structures are further poised to help inform on the mechanism of activation of the stable metazoan γTuRC complexes. Both of the published γTuRC structures would require large conformational changes in pitch and rise to match MT symmetry (*Figure 7*). In the human γTuRC, the γTuNA CM1 helix is bound at the GCP2:GCP6 interface, and the distance between the GCPs at this interface closely matches that observed in our γTuRC structures, indicating a conserved and more optimal spacing upon CM1 binding (*Figure 7—figure supplements 1* and *2B*). This suggests that the mode of interaction of the Spc110p$^{CM1}$ helix with GCPs is broadly conserved. The fact that both structures have low-resolution density extending past the CM1 N-terminus towards the adjacent GCP suggests that there may also be a conserved functional role for the residues N-terminal to CM1 (*Figure 7—figure supplement 1BC*).

Simple addition of a CDK5RAP2 homologue during purification did not yield observable CM1 density in the *Xenopus* γTuRC complexes (*Liu et al., 2020*), suggesting either a lower affinity for the other sites or that other factors could be important. Combined with our additional observation of a change in the local twist and pitch of GCP:γ-tubulin conformation near GCP2:GCP6, the data

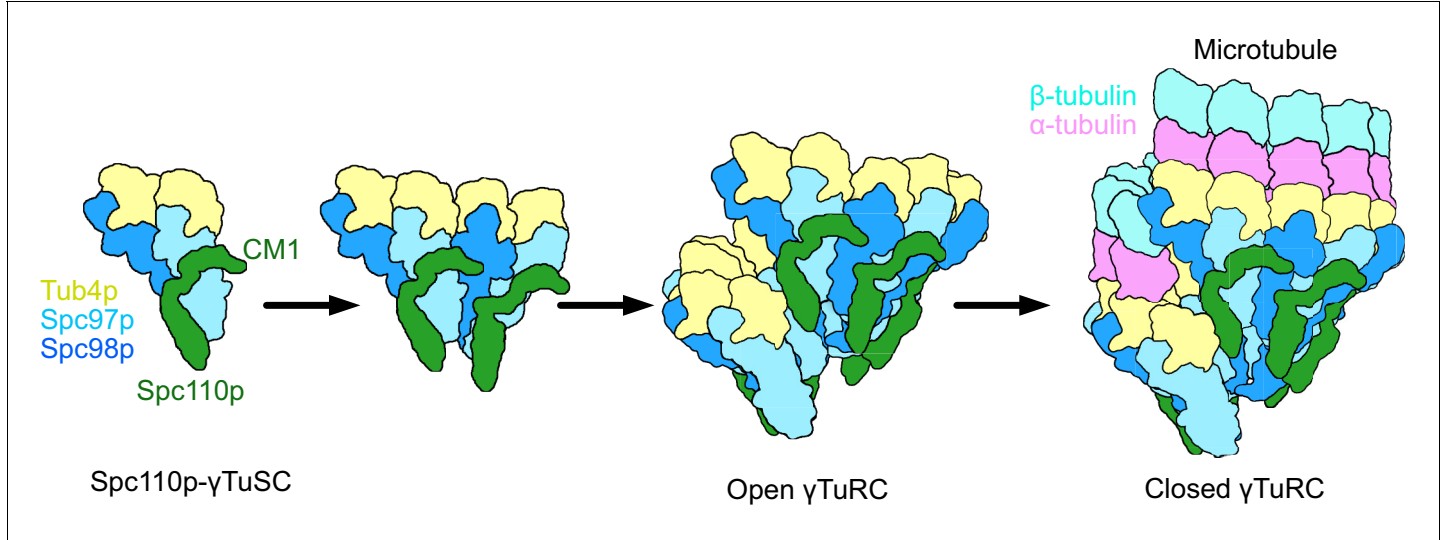

**Figure 8.** Model of γ-tubulin ring complex (γTuRC) assembly and activation. γ-Tubulin small complex (γTuSC) monomers bound to Spc110p display an improved binding for γTuSC due to the presence of the overhanging Spc110p$^{CM1}$ binding surface. This leads to cooperative assembly of further γTuSCs to form an open γTuRC. The open γTuRC then transitions into a closed structure either prior to or concurrent with microtubule nucleation. Only six full Spc110p$^{CM1}$ binding sites exist in a fully formed γTuRC, matching the hexagonal Spc42p symmetry at the spindle pole body (**Bullitt et al., 1997**; **Drennan et al., 2019**; **Muller et al., 2005**).

suggests that binding of a CM1 helix at the five GCP2:GCP3 inter-γTuSC interfaces could cooperatively rearrange the γTuRC to much better match the MT pitch and spacing, leading to activation of MT nucleation.

Recent work on MT nucleation from single purified metazoan γTuRCs has suggested that MT nucleation remains a highly cooperative process, requiring ~4–7 αβ-tubulin dimers (**Consolati et al., 2020**; **Thawani et al., 2020**). An optimal nucleator that perfectly matches the MT symmetry would be expected to exhibit non-cooperative behavior, as is observed in elongating MTs. In metazoans, factors that allosterically drive a more ideal template state could reduce the energetic barrier to nucleation and stimulate MT nucleation.

Consistent with this interpretation, CDK5RAP2 has been found to stimulate MT nucleation in vitro (**Choi et al., 2010**). This further suggests a functional role for increasing the local concentration of CM1-containing proteins through either an ordered oligomerization processes, as with Spc110p (**Lyon et al., 2016**), or through a more colligative phase-condensate mechanism. For example, if the additional CM1s were to come from proteins tightly bound within the PCM (such as pericentrin, centrosomin, or CDK5RAP2), the effect would be to couple γTuRC activation to PCM localization, similar to Spc110p confining yeast γTuRC function to the SPB. Despite these major advances, significant gaps remain in our understanding of how the binding of regulatory proteins and PTMs acts to modulate activation of yeast and metazoan γTuRCs.

## Data deposition

Structure factors and model coordinates for the Xrcc4-Spc110p$^{164-207}$ fusion X-ray crystal structure have been uploaded to the RCSB Protein Data Bank with PDB ID 7M3P.

Cryo-EM reconstructions and model coordinates have been deposited to the EMDB and PDB for the γTuSC monomer (EMDB ID: EMD-23638; PDB ID: 7M2Z), γTuRC$^{SS}$ (EMDB ID: EMD-23635; PDB ID: 7M2W), γTuRC$^{WT}$ open (EMDB ID: EMD-23636; PDB ID: 7M2X), and γTuRC$^{WT}$ closed (EMDB ID: EMD-23637; PDB ID: 7M2Y) states. The cryo-EM reconstruction for the γTuSC dimer (EMDB ID: EMD-23639) has been deposited to the EMDB. Accession codes are also available in **Supplementary files 1** and **2**.

XL-MS experiments and data analysis are described in the Materials and methods section. All raw and processed data, along with complete information required to repeat the current analyses, can be found at https://proxl.yeastrc.org/proxl/p/cm1-tusc as described in the Materials and methods

section. In addition, the complete crosslinking dataset and analysis presented in this paper can be viewed, downloaded, examined, and visualized using our web-based interface, ProXL, at the URL above.

Integrative modeling scripts and final models and densities are available at https://salilab.org/gtuscSpc110 and have been deposited to the Protein Data Bank archive for integrative structures (https://pdb-dev.wwpdb.org/) with depositions codes PDBDEV_00000077, PDBDEV_00000078, and PDBDEV_00000079.

# Materials and methods

## Key resources table

| Reagent type (species) or resource | Designation | Source or reference | Identifiers | Additional information |
|---|---|---|---|---|
| Software, algorithm | IMP (Integrative Modeling Platform) | https://integrativemodeling.org; https://doi.org/10.1371/journal.pbio.1001244 | RRID:SCR_002982 | Version 2.8 |
| Software, algorithm | UCSF Chimera | https://www.cgl.ucsf.edu/chimera/ https://doi.org/10.1002/jcc.20084 | RRID:SCR_004097 | |
| Strain, strain background (*Escherichia coli*) | BL21(DE3) CodonPlus-RIL | Agilent | Part No.:230245 | |
| Genetic reagent (*Homo sapiens*, *Saccharomyces cerevisiae*) | pET28a-3C-Xrcc4-Spc110(164-207) | This paper | Uniprot:Q13426 (Xrcc4); Uniprot:32380 (Spc110) | Construct contains residues 2–132 of *H. sapiens* Xrcc4 fused with residues 164–207 of *S. cerevisiae* Spc110 |
| Software, algorithm | XDS | *Kabsch, 2010*; DOI: https://doi.org/10.1107/S0907444909047337 | RRID:SCR_015652 | Version: October 15, 2015 |
| Software, algorithm | Phenix | *Adams et al., 2010*; DOI: https://doi.org/10.1107/S0907444909052925; *McCoy et al., 2007*; DOI: https://doi.org/10.1107/S0021889807021206; *Terwilliger et al., 2008*; DOI: https://doi.org/10.1107/S090744490705024X *Afonine et al., 2012*; DOI: https://doi.org/10.1107/S0907444912001308 | RRID:SCR_014224 | Version 1.10.1_2155 |
| Software, algorithm | Coot | *Emsley et al., 2010*; DOI: https://doi.org/10.1107/S0907444910007493 | RRID:SCR_014222 | Version 0.8.3 |
| Software, algorithm | Kojak, XL identification algorithm | http://www.kojak-ms.org/ | RRID:SCR_021028 | Versions 1.4.1 and 1.4.3 |
| Software, algorithm | ProXL, protein XL data visualization | https://proxl-ms.org/ | RRID:SCR_021027 | |
| Chemical compound, drug | DSS | Thermo Fisher Scientific | 21655 | |
| Chemical compound, drug | EDC | Thermo Fisher Scientific | A35391 | |
| Chemical compound, drug | Sulfo-NHS | Thermo Fisher Scientific | A39269 | |
| Software, algorithm | cisTEM | *Grant et al., 2018*. DOI:10.7554/eLife.35383 | RRID:SCR_016502 | Version 1.0 beta |
| Software, algorithm | Relion | *Scheres, 2012*. PMID:23000701; DOI: 10.1016/j.jsb.2012.09.006 | RRID:SCR_016274 | |
| Genetic reagent (*S. cerevisiae*) | pFastBac-Tub4p | *Vinh et al., 2002*. doi: 10.1091/mbc.02-01-0607 | | |
| Genetic reagent (*S. cerevisiae*) | pFastBac-Spc97p | *Vinh et al., 2002*. doi: 10.1091/mbc.02-01-0607 | | |

*Continued on next page*

*Continued*

| Reagent type (species) or resource | Designation | Source or reference | Identifiers | Additional information |
|---|---|---|---|---|
| Genetic reagent (*S. cerevisiae*) | pFastBac-Spc98p | *Vinh et al., 2002*. doi: 10.1091/mbc.02-01-0607 | | |
| Genetic reagent (*S. cerevisiae*) | pFastBac-GST-Spc110p$^{1-220}$ | *Vinh et al., 2002*. doi: 10.1091/mbc.02-01-0607 | | |
| Genetic reagent (*S. cerevisiae*) | pFastBac-Tub4p$^{S58C/G288C}$ | *Kollman et al., 2015*. DOI: 10.1038/nsmb.2953 | | |

## γTuSC purification

γTuSC was prepared essentially as described (*Vinh et al., 2002*; *Lyon et al., 2016*).

## Crosslinking and mass spectrometry (XL-MS)

XL-MS was carried out as described by *Zelter et al., 2015*. All γTuSC-Spc110p reactions were in 40 mM HEPES pH 7.0, 150 mM KCl, and contained a final concentration 0.4 µM γTuSC and 0.8 µM Spc110. DSS reactions were carried out at room temperature (RT) for 3 min using 0.44 mM DSS prior to quenching with 100 mM ammonium bicarbonate. EDC reactions were carried out at RT for 30 min using 5.4 mM EDC plus 2.7 mM Sulfo-NHS prior to quenching with 100 mM ammonium bicarbonate plus 20 mM 2-mercaptoethanol. After quenching, reactions were reduced for 30 min at 37°C with 10 mM dithiothreitol (DTT) and alkylated for 30 min at RT with 15 mM iodoacetamide. Trypsin digestion was performed at 37°C for 4 or 6 hr with shaking at a substrate to enzyme ratio of 17:1 or 30:1 for EDC and DSS reactions, respectively, prior to acidification with 5 M HCl. Digested samples were stored at −80°C until analysis. Mass spectrometry and data analysis were performed as described (*Zelter et al., 2015*). In brief, 0.25 µg of sample was loaded onto a fused-silica capillary tip column (75 µm i.d.) packed with 30 cm of Reprosil-Pur C18-AQ (3 µm bead diameter, Dr. Maisch) and eluted at 0.25 µL/min using an acetonitrile gradient. Mass spectrometry was performed on a QExactive HF (Thermo Fisher Scientific) in a data-dependent mode and spectra converted to mzML using msconvert from ProteoWizard (*Chambers et al., 2012*).

Proteins present in the sample were identified using Comet (*Eng et al., 2013*). Crosslinked peptides were identified within those proteins using Kojak version 1.4.1 or 1.4.3 (*Hoopmann et al., 2015*) available at http://www.kojak-ms.org. Percolator version 2.08 (*Käll et al., 2007*) was used to assign a statistically meaningful *q* value to each peptide spectrum match (PSM) through analysis of the target and decoy PSM distributions. Target databases consisted of all proteins identified in the sample analyzed. Decoy databases consisted of the corresponding set of reversed protein sequences. Data were filtered to show hits to the target proteins that had a Percolator assigned peptide level *q* value ≤ 0.01 and a minimum of two PSMs. The complete list of all PSMs and their Percolator assigned *q* values is available on the ProXL web application (*Riffle et al., 2016*) at https://proxl.yeastrc.org/proxl/p/cm1-tusc along with the raw MS spectra and search parameters used.

## Xrcc4-Spc110$^{164-207}$ purification and X-ray crystallography

DNA encoding residues 2–132 of *Homo sapiens* Xrcc4 (UniProt ID Q13426) fused in frame with residues 164–207 of Spc110p were synthesized by GeneArt (Thermo Fisher Scientific) and cloned into pET28a expression vector with N-terminal 6His tag, 3C protease cleavage site, and six-residue linker with sequence GSGGSG. Xrcc4-Spc110$^{164-207}$ was expressed in *Escherichia coli* BL21-CodonPlus-RIL (Agilent). Cells were harvested by centrifugation, then resuspended in lysis buffer (50 mM potassium phosphate pH 8, 300 mM NaCl, 5 mM EDTA, 1 mM DTT, 0.3% Tween-20, 1x cOmplete protease inhibitor, EDTA-free [Roche]). Cells were lysed by Emulsiflex C3 (Avestin). Lysate was cleared by ultracentrifugation at 40,000 x g for 30 min in a Type 45Ti rotor (Beckman-Coulter). Xrcc4-Spc110$^{164-207}$ was then purified by NiNTA affinity chromatography followed by addition of 3C protease overnight at 4°C to cleave the 6His tag. Xrcc4-Spc110$^{164-207}$ was further purified by size exclusion chromatography (Superdex 75; GE Healthcare Life Sciences), anion exchange chromatography (MonoQ; GE Healthcare Life Sciences), with a final size exclusion polishing and buffer exchange step (Superdex 75). Crystals of Xrcc4-Spc110$^{164-207}$ were obtained by hanging drop vapor diffusion with 8 mg/mL protein and a well solution containing 13% PEG3350 and 0.2 M magnesium formate. Crystals

were cryo-protected by rapid transfer to well solution with 30% PEG3350. Diffraction data was collected under cryogenic conditions at Advanced Light Source beamline 8.3.1. Diffraction data was processed with XDS (*Kabsch, 2010*) and indexed in space group P1. Phases were obtained by molecular replacement using Phaser within the Phenix package (*Adams et al., 2010*; *McCoy et al., 2007*). The search model was the PDB ID 1FU1 residues 1–150, with the coiled-coil residues 133–150 mutated to alanine. The S-(dimethylarsenic)cysteine at position 130 in 1FU1 was modified to cysteine. The majority of the structure was built with phenix.autobuild (*Terwilliger et al., 2008*) with the remainder built manually in Coot (*Emsley et al., 2010*) and refined with phenix.refine (*Afonine et al., 2012*). The final structure contains Spc110 residues 164–203, along with the Xrcc4 fusion domain.

## Filament purification

Filaments were prepared essentially as described (*Kollman et al., 2010*; *Kollman et al., 2015*) with slight modifications.

The buffer used during purification was modified to contain 40 mM HEPES pH 7.5, 100 mM KCl, 1 mM EGTA, 2 mM MgCl2, 10% glycerol, and 1 mM DTT. Samples were concentrated and buffer exchanged to obtain a final glycerol concentration of 2.5% glycerol.

Oxidation of $\gamma$TuSC$^{SS}$ filaments was performed overnight at 4°C by dialysis into 1 mM oxidized glutathione, removing DTT.

## Grid preparation: $\gamma$TuSC

Prior to grid preparation, $\gamma$TuSC aliquots were centrifuged in a benchtop centrifuge (Eppendorf 5415D) at 16,000 g for 15 min and transferred to a new tube. The sample concentration was assessed on a nanodrop and diluted to a final concentration of ~1 $\mu$M (O.D. at 280 nm wavelength of 0.28–0.35) such that the final buffer conditions were 40 mM HEPES pH 7.5, 2 mM MgCl$_2$, 1 mM EGTA, 1 mM GDP, 100 mM KCl, and 2.5% v/v glycerol.

Data used for initial model generation and refinement had final buffer conditions of 40 mM HEPES pH 7.5, 1 mM MgCl$_2$, 1 mM EGTA, and 100 mM KCl.

C-flat 1.2–1.3 4C grids were used for sample freezing and glow discharged for ~30 s at −20 mA immediately prior to plunge-freezing. Grids were frozen on a Vitrobot Mark II or Mark IV, with the humidity set to 100%, and using Whatman 1 55-mm filter papers.

## Grid preparation: $\gamma$TuRC filaments

Quantifoil 1.2–1.3 400-mesh grids were used for sample freezing and glow discharged for ~30 s at −20 mA immediately prior to plunge-freezing. Grids were frozen on a Vitrobot Mark IV, with the humidity set to 100%, and using Whatman 1 55-mm filter papers.

The final conditions used for $\gamma$TuRC$^{WT}$ filament freezing were 40 mM HEPES pH 7.5, 2 mM MgCl$_2$, 1 mM EGTA, 1 mM GDP, 100 mM KCl, 1 mM DTT, and 2.5% v/v glycerol.

The final conditions used for $\gamma$TuRC$^{SS}$ filament freezing were 40 mM HEPES pH 7.5, 2 mM MgCl$_2$, 1 mM EGTA, 0.5 mM GTP, 100 mM KCl, 1 mM oxidized glutathione, and 2.5% v/v glycerol.

## Electron microscopy: $\gamma$TuSC single-particle data

Micrographs used in $\gamma$TuSC initial model generation were collected using an FEI Tecnai F20 operated at 200 kV at a nominal magnification of 29,000× (40,322× at the detector). The data was collected with a 20 $\mu$m C2 aperture and a 100 $\mu$m objective aperture with a target underfocus of ~1–2.5 $\mu$m. UCSF Image4 (*Li et al., 2015*) was used to operate the microscope. Dose-fractionated micrographs were collected on a Gatan K2 Summit camera in super-resolution mode at a dose rate of ~8.5–9.5 electrons per physical pixel per second for 12 s, with the dose fractionated into 40 frames.

Micrographs included in the final model were collected using an FEI Tecnai Polara operated at 300 kV at a nominal magnification of 31,000× (39,891× at the detector). Data was collected with a 30 $\mu$m C2 aperture and a 100 $\mu$m objective aperture inserted with a target underfocus of ~1–3 $\mu$m. Leginon (*Suloway et al., 2005*) or SerialEM (*Mastronarde, 2005*) were used to operate the microscope. Dose-fractionated micrographs were collected on a Gatan K2 Summit camera in super-

resolution mode at a dose rate of approximately six electrons per physical pixel per second for 20 s, with the dose fractionated into 100 frames.

## Electron microscopy: γTuRC^WT filament data

Data was collected in two sessions on a Titan Krios operated at 300 kV at a nominal magnification of 22,500× (47,214× at the detector). The data was collected with a 70 μm C2 aperture and a 100 μm objective aperture with a target underfocus of ~0.9–2.0 μm. Dose-fractionated micrographs were collected on a Gatan K2 Summit camera in super-resolution mode at a dose rate of six electrons per physical pixel per second for 15 s, with the dose fractionated into 75 frames.

## Electron microscopy: γTuRC^SS filament data

Data was collected in two sessions on a Titan Krios operated at 300 kV at a nominal magnification of 22,500× (47,214× at the detector). The data was collected with a 70 μm C2 aperture and a 100 μm objective aperture with a target underfocus of ~0.6–2.0 μm. Dose-fractionated micrographs were collected on a Gatan K2 Summit camera in super-resolution mode at a dose rate of 6.7 electrons per physical pixel per second for 12 s, with the dose fractionated into 120 frames.

## Image processing: γTuSC single-particle initial model generation

Dose-fractionated image stacks were corrected for drift and beam-induced motion as well as binned twofold from the super-resolution images using MotionCorr (*Li et al., 2013*). CTF estimation was performed using CTFFIND4 (*Rohou and Grigorieff, 2015*). Particle coordinates were semi-automatically picked from filtered and binned images using the e2boxer 'swarm' tool (*Tang et al., 2007*). Particles were extracted using Relion (*Scheres, 2012*) with a box size of 384 physical pixels resampled to 96 pixels for initial processing. A dataset of ~50,000,000 particles from 217 micrographs was used to generate 300 2D classes using Relion 1.3. 23 classes were selected and used in the generation of a γTuSC monomer initial model using the e2initialmodel.py function in EMAN2. This model was then used as a reference in Relion 1.3 for 3D classification into four classes of a 115,701 particle dataset from 507 micrographs with a 384 pixel box. Particles from the best γTuSC monomer class were then used for further processing and classification into four classes in FREALIGN (*Grigorieff, 2016*). The best class, with a resolution of ~9 Å, was then used as a 3D reference for processing of the Polara data.

## Image processing: γTuSC single-particle Polara data

Images were drift-corrected and dose-weighted using MotionCor2 (*Zheng et al., 2017*). Initial processing to generate monomer and dimer reconstructions was performed with CTFFIND4 (*Rohou and Grigorieff, 2015*), Relion (*Scheres, 2012*), and FREALIGN (*Grigorieff, 2016*). Processing leading to the final reconstructions was performed in cisTEM (*Grant et al., 2018*). Particles were automatically picked from 7381 images in cisTEM, yielding 3,210,917 initial particle coordinates. 2D classification was performed to eliminate junk and ice particles, with 1,187,292 particles being included in the initial 3D classification. During 3D classification, particles were extracted from unbinned super-resolution micrographs with a box size of 376.02 Å (600 pixels).

Classification and alignment were performed using the cisTEM 'Manual Refine' tool, as delineated in Figure S13.

## Image processing: γTuRC^WT filaments

Images were drift-corrected, dose-weighted, and binned twofold using MotionCor2 (*Zheng et al., 2017*). Filaments were manually picked using e2helixboxer (*Tang et al., 2007*) from 2204 micrographs. Filaments were extracted in Relion2 (*Kimanius et al., 2016*) and boxed approximately every three asymmetric units, using a rise of 21 Å with a box size of 635.4 Å (600 pixels on the micrographs, rescaled to 448 pixels), yielding 28,753 boxed filament images. 2D classification was performed to eliminate junk particles and filament ends, with 28,648 filament images remaining after culling. These images were initially aligned in Relion2 (*Kimanius et al., 2016*, p. 2), while allowing for the refinement of helical parameters. Particle alignments were exported into FREALIGN (*Grigorieff, 2016*) for additional helical refinement. FREALIGN alignments were used for helical symmetry expansion as implemented in Relion2. Symmetry-expanded alignment parameters were then

imported into cisTEM (*Grant et al., 2018*) for local alignment and classification. A user-generated mask was supplied for these refinements, with the final mask containing approximately three γTuSC subunits with a total molecular weight of approximately 900 kDa. Prior to classification, the defocus was refined in cisTEM. Focused classification was performed in cisTEM, as delineated in *Figure 3—figure supplement 1*.

### Image processing: γTuRC<sup>SS</sup> filaments

Images were drift-corrected, dose-weighted, and binned twofold using MotionCor2 (*Zheng et al., 2017*). Filaments were manually picked using e2helixboxer from 3024 micrographs. Filaments were extracted in Relion2 (*Kimanius et al., 2016*) and boxed approximately every three asymmetric units, using a rise of 21 Å with a box size of 635.4 Å (600 pixels), yielding 175,500 boxed filament images. 2D classification was performed to eliminate junk particles and filament ends, with 152,798 filament images remaining after culling. These images were initially aligned in Relion2, while allowing for the refinement of helical parameters. Particle alignments were exported into FREALIGN (*Grigorieff, 2016*) for additional helical refinement. FREALIGN alignments were used for helical symmetry expansion as implemented in Relion2. Symmetry-expanded alignment parameters were then imported into cisTEM (*Grant et al., 2018*) for local alignment and classification. A user-generated mask was supplied for these refinements, with the final mask containing approximately three γTuSC subunits with a total molecular weight of approximately 900 kDa. Prior to classification, the defocus was refined in cisTEM. Focused classification was performed in cisTEM, as delineated in *Figure 3—figure supplement 2*.

### Difference map generation

The γTuRC<sup>SS</sup> reconstruction was resampled to 400 pixels using resample.exe included in the cisTEM package. A molecular map of a trimer of γTuSCs from the γTuRC<sup>SS</sup> model, but not including Spc110p, was generated in Chimera using the molmap command with a resolution of 3.3 Å. A difference map was generated using the diffmap.exe software obtained from the Grigorieff lab website (https://grigoriefflab.umassmed.edu/diffmap). The difference map was sharpened with a b-factor of −40 Å$^2$ and filtered to 5.5 Å with a five-pixel fall-off using the bfactor software obtained from the Grigorieff lab website (https://grigoriefflab.umassmed.edu/bfactor).

### Local resolution estimation

Local resolutions were estimated in blocres (*Heymann, 2001*) using a box size of 20 and a step of either 1 (γTuRC<sup>SS</sup>, γTuRC<sup>WT</sup>) or 2 (monomer and dimer). The local resolution estimate was applied to the γTuRC<sup>SS</sup> reconstruction using SPOC in *Figure 3—figure supplement 3C* (*Beckers and Sachse, 2020*).

### Initial atomic model generation: γTuSC monomers

To generate an initial atomic model, the crystal structure of human GCP4 and a previously generated pseudo-atomic model were used as templates. Prior to fitting, the GCP4 structure was threaded with the Spc97p and Spc98p sequence, and the human γ-tubulin was threaded with the Tub4p sequence. These initial models were fitted into preliminary structures into segmented density using Rosetta's relax function. Missing residues were built using RosettaCM density-guided model building (*DiMaio et al., 2015*), with the human GCP4, γ-tubulin threaded models and the pseudo-atomic model being sampled separately. Well-scoring structures were then compared to the density, assessing the quality of the fit to determine the register. In cases where the register was poorly fit and the correct register was clear, the register was manually adjusted to fit map details. Certain regions were built using the RosettaES algorithm (*Frenz et al., 2017*). This procedure was iterated, with occasional manual modification of the structure in Coot (*Emsley et al., 2010*).

As a final step, final half-maps were used in the refinement, with the best preliminary models relaxed and refined through iterative backbone rebuilding (*Wang et al., 2016*) into one half-map reconstruction, and iteratively refined using Rosetta. This model was used as a starting point for atomic model building into the higher resolution γTuSC<sup>SS</sup> filament structure.

## Atomic model generation: γTuRC<sup>SS</sup>

The initial model from monomer fitting was relaxed into the γTuRC<sup>SS</sup> structure using Rosetta's relax function and refined using iterative backbone rebuilding as previously described. Poorly fitting and missing regions were either built in Coot or using the RosettaES algorithm. Residues Spc110p$^{112\text{-}150}$ were manually built in Coot. Finally, the models were iteratively refined using a procedure that involved using Rosetta to relax the models into one half-map and iterative backbone rebuilding, with the best models as assessed using the FSC to the second half-map being combined using the phenix combine_models function, followed by Phenix (*Adams et al., 2010*) real-space refinement (*Afonine et al., 2018*) and manual modification. This model was used as the basis for the single-particle monomer model, and the γTuRC<sup>WT</sup> models. Models were further iteratively refined using Rosetta, Coot, and Phenix. Finally, the Spc110p$^{164\text{-}208}$ crystal structure was relaxed into γTuRC<sup>SS</sup> density, with the residues 151–164 built manually in Coot. Spc110p was iteratively relaxed into density using Rosetta to relax the models into one half-map and iterative backbone rebuilding, with the best models being visually inspected and manually modified in Coot. A final round of manual refinement of Spc110p was performed in ISOLDE (*Croll, 2018*) using a density-modified map generated in Phenix (*Terwilliger et al., 2020*).

## Atomic model generation: γTuRC<sup>WT</sup> open state

The initial model from γTuSC<sup>SS</sup> fitting was relaxed into the γTuSC<sup>WT</sup> structure using Rosetta's relax function and refined using iterative backbone rebuilding, with the best models as assessed using the FSC to the second half-map being combined using the phenix combine_models function. Models were further iteratively refined using Rosetta, Coot, and Phenix.

## Atomic model generation: γTuRC<sup>WT</sup> closed state

The initial model from γTuSC<sup>SS</sup> fitting was relaxed into the closed γTuSC<sup>WT</sup> structure using Rosetta's relax function and refined using iterative backbone rebuilding, with the best models as assessed using the FSC to the second half-map being combined using the phenix combine_models function, followed by Phenix real-space refinement and manual modification. Models were further iteratively refined using Rosetta, Coot, and Phenix.

## Model generation: γTuSC monomer

The initial model from γTuSC<sup>SS</sup> fitting was relaxed into the γTuSC monomer structure using Rosetta's relax function and refined using iterative backbone rebuilding, with the best models as assessed using the FSC to the second half-map being combined using the phenix combine_models function, followed by Phenix real-space refinement and manual modification. Models were further iteratively refined using Rosetta, Coot, and Phenix. The final round of Phenix real-space refinement was performed against the full map.

## Model generation: γTuSC dimer

The dimer model was generated by using Rosetta's relax function to fit two γTuSC<sup>SS</sup> models generated as above into dimer density. The nucleotide was subsequently modified to GDP, and poorly fitting regions were deleted. This model was used solely for the segmentation shown in *Figure 5—figure supplement 3*.

## Surface area calculations

Surface area calculations were performed using NACCESS (*Hubbard and Thornton, 1993*).

## 2D classification: Figure S4—figure supplement 1

Monomer and dimer stacks (384 pixel stacks used in final reconstruction generation) were separately classified using cisTEM (*Grant et al., 2018*). Classes showing high-resolution features were extracted for figure generation using IMOD (*Kremer et al., 1996*).

## Wiring diagrams

Wiring diagrams were generated using the PDBSum online portal (*Laskowski, 2009*).

## Sequence alignments for conservation surfaces

Sequences for Spc97p and Spc98p homologues from *H. sapiens*, *Mus musculus*, *Danio rerio*, *Xenopus laevis*, *Drosophila melanogaster*, *Arabidopsis thaliana*, *Glycine max*, *Dictyostelium discoideum*, and *Saccharomyces pombe* were aligned to the sequence from *Saccharomyces cerevisiae* using the MAFFT algorithm (*Katoh and Standley, 2013*) implemented on the MPI bioinformatics website (*Zimmermann et al., 2018*). The Spc110p sequence from *S. cerevisiae* was similarly aligned to orthologs from *H. sapiens*, *M. musculus*, *D. rerio*, *X. laevis*, *D. melanogaster*, *D. discoideum*, and *S. pombe*. Sequence alignments were imported using the Multalign Viewer in Chimera (*Pettersen et al., 2004*), which was subsequently used to color the surfaces and ribbons by conservation.

## Figure generation

Structural figures were generated in UCSF Chimera (*Pettersen et al., 2004*) or ChimeraX (*Goddard et al., 2018*). FSC plots were generated in Excel from Part_FSC estimates in cisTEM (*Grant et al., 2018*). Map-to-model FSCs were generated in Phenix (*Adams et al., 2010*), with default parameters. Figures panels were compiled into figures in Affinity Designer.

## Acknowledgements

We gratefully acknowledge many helpful discussions with members of the Agard lab, as well as with our collaborators on the Yeast Centrosome – Structure, Assembly, and Function program project grant in the labs of Mark Winey (PI), Trisha Davis, Chip Asbury, Ivan Rayment, Andrej Sali, and Sue Jaspersen. We would like to thank Michael Braunfeld, Alexander Myasnikov, and David Bulkley for their work running the electron microscopy facility at UCSF; Cameron Kennedy, Matthew Harrington, and Joshua Baker-Lepain for their work running the UCSF MSG and wynton HPC clusters; Richard Johnson for assisting with MS data acquisition; Michelle Moritz for help and training with the purification of γTuSC; and Ray Yu-Ruei Wang for advice with modeling in Rosetta. We acknowledge financial support from the following sources: Howard Hughes Medical Institute (DAA), National Institute of General Medical Sciences (NIGMS) grants: R01 GM031627 (DAA), R35GM118099 (DAA), and P01 GM105537 (DAA, TD, AS), P41 GM103533 (TD, MM), GM083960 (AS), GM109824 (AS), NSF Graduate Research Fellowship 1144247 (AL), and UCSF Discovery Fellowship (AL). Also, NIH S10 support for UCSF cryo-EM and computing (1S10OD020054, 1S10OD021741). Beamline 8.3.1 at the Advanced Light Source is operated by the University of California Office of the President, Multicampus Research Programs and Initiatives grant MR-15–328599, the National Institutes of Health (R01 GM124149 and P30 GM124169), Plexxikon Inc, and the Integrated Diffraction Analysis Technologies program of the US Department of Energy Office of Biological and Environmental Research. The Advanced Light Source (Berkeley, CA) is a national user facility operated by Lawrence Berkeley National Laboratory on behalf of the US Department of Energy under contract number DE-AC02-05CH11231, Office of Basic Energy Sciences.

## Additional information

### Funding

| Funder | Grant reference number | Author |
| --- | --- | --- |
| Howard Hughes Medical Institute | 0714 | David A Agard |
| National Institute of General Medical Sciences | GM031627 | David A Agard |
| National Institute of General Medical Sciences | GM118099 | David A Agard |
| National Institute of General Medical Sciences | GM105537 | Andrej Sali<br>Trisha N Davis<br>David A Agard |
| National Institute of General Medical Sciences | GM103533 | Michael J MacCoss<br>Trisha N Davis |

| National Institute of General Medical Sciences | GM083960 | Andrej Sali |
| --- | --- | --- |
| National Institute of General Medical Sciences | GM109824 | Andrej Sali |
| National Science Foundation | 1144247 | Andrew S Lyon |
| UCSF Foundation | 2014 | Andrew S Lyon |
| National Institute of General Medical Sciences | 1S10OD020054 | David A Agard |
| National Institute of General Medical Sciences | 1S10OD021741 | David A Agard |

The funders had no role in study design, data collection and interpretation, or the decision to submit the work for publication.

## Author contributions

Axel F Brilot, Resources, Data curation, Formal analysis, Validation, Investigation, Visualization, Methodology, Writing - original draft, Writing - review and editing, AB purified proteins complexes and optimized sample preparation for cryo-EM, performed EM imaging experiments, cryo-EM image analysis and built atomic models; Andrew S Lyon, Resources, Data curation, Formal analysis, Funding acquisition, Validation, Investigation, Visualization, Methodology, Writing - original draft, Writing - review and editing, Created expression constructs, Purified proteins, Performed biochemical analyses; Alex Zelter, Data curation, Formal analysis, Validation, Investigation, Writing - original draft, Project administration, Writing - review and editing, Performed protein crosslinking and mass spectrometry; Shruthi Viswanath, Software, Formal analysis, Supervision, Funding acquisition, Validation, Investigation, Visualization, Methodology, Writing - original draft, Writing - review and editing, Performed integrative modeling; Alison Maxwell, Resources, Supervision, Funding acquisition, Investigation, Writing - review and editing, Created expression constructs, Purified proteins, Performed biochemical analyses; Michael J MacCoss, Supervision, Funding acquisition; Eric G Muller, Supervision, Funding acquisition, Project administration, Writing - review and editing; Andrej Sali, Resources, Supervision, Funding acquisition, Investigation, Project administration, Writing - review and editing; Trisha N Davis, David A Agard, Conceptualization, Resources, Supervision, Funding acquisition, Investigation, Project administration, Writing - review and editing

## Author ORCIDs

Axel F Brilot  https://orcid.org/0000-0001-8548-4224
Andrew S Lyon  https://orcid.org/0000-0001-7681-4981
Michael J MacCoss  http://orcid.org/0000-0003-1853-0256
Andrej Sali  http://orcid.org/0000-0003-0435-6197
Trisha N Davis  http://orcid.org/0000-0003-4797-3152
David A Agard  https://orcid.org/0000-0003-3512-695X

## Decision letter and Author response

Decision letter https://doi.org/10.7554/eLife.65168.sa1
Author response https://doi.org/10.7554/eLife.65168.sa2

# Additional files

## Supplementary files

- Supplementary file 1. X-ray data collection and refinement statistics.
- Supplementary file 2. Cryo-EM data collection, processing, and modeling statistics.
- Transparent reporting form

## Data availability

Structure factors and model coordinates for the Xrcc4-Spc110p164-207 fusion X-ray crystal structure have been uploaded to the RCSB protein data bank with PDB ID 7M3P. Cryo-EM reconstructions and model coordinates have been deposited to the EMDB and PDB for the γTuSC monomer (EMDB ID: EMD-23638 PDB ID: 7M2Z), γTuRCSS (EMDB ID: EMD-23635 PDB ID: 7M2W), γTuRCWT open (EMDB ID: EMD-23636 PDB ID: 7M2X) and γTuRCWT closed (EMDB ID: EMD-23637 PDB ID: 7M2Y) states. The cryo-EM reconstruction for the γTuSC dimer (EMDB ID: EMD-23639) has been deposited to the EMDB. Accession codes are also available in Tables S1 and S2. XL-MS experiments and data analysis are described in the Methods section. All raw and processed data, along with complete information required to repeat the current analyses, can be found at https://proxl.yeastrc.org/proxl/p/cm1-tusc as described in the Methods section. In addition, the complete crosslinking dataset and analysis presented in this paper can be viewed, downloaded, examined and visualized using our web-based interface, ProXL, at the URL above. Integrative modeling scripts and final models and densities are available at https://salilab.org/gtuscSpc110 and have been deposited to the Protein Data Bank archive for integrative structures (https://pdb-dev.wwpdb.org/) with depositions codes PDBDEV_00000077 PDBDEV_00000078 PDBDEV_00000079.

The following datasets were generated:

| Author(s) | Year | Dataset title | Dataset URL | Database and Identifier |
|---|---|---|---|---|
| Brilot AF, Lyon AS, Zelter A, Viswanath S, Maxwell A, MacCoss MJ, Muller EG, Sali A, Davis TN, Agard DA | 2021 | XRCC4-Spc110p(164-207) fusion x-ray | https://www.rcsb.org/structure/7m3p | RCSB Protein Data Bank, 7M3P |
| Brilot AF, Lyon AS, Zelter A, Viswanath S, Maxwell A, MacCoss MJ, Muller EG, Sali A, Davis TN, Agard DA | 2021 | XL-MS | https://proxl.yeastrc.org/proxl/p/cm1-tusc | proxl Protein Cross-Linking Database, https://proxl.yeastrc.org/proxl/viewProject.do?project_id=86 |
| Brilot AF, Lyon AS, Zelter A, Viswanath S, Maxwell A, MacCoss MJ, Muller EG, Sali A, Davis TN, Agard DA | 2021 | gTuSC monomer model | https://www.rcsb.org/structure/7M2Z | RCSB Protein Data Bank, 7M2Z |
| Brilot AF, Lyon AS, Zelter A, Viswanath S, Maxwell A, MacCoss MJ, Muller EG, Sali A, Davis TN, Agard DA | 2021 | gTuRC(SS) model | https://www.rcsb.org/structure/7M2W | RCSB Protein Data Bank, 7M2W |
| Brilot AF, Lyon AS, Zelter A, Viswanath S, Maxwell A, MacCoss MJ, Muller EG, Sali A, Davis TN, Agard DA | 2021 | gTuRC(WT) closed model | https://www.rcsb.org/structure/7M2Y | RCSB Protein Data Bank, 7M2Y |
| Brilot AF, Lyon AS, Zelter A, Viswanath S, Maxwell A, MacCoss MJ, Muller EG, Sali A, Davis TN, Agard DA | 2021 | gTuRC(WT) open model | https://www.rcsb.org/structure/7M2X | RCSB Protein Data Bank, 7M2X |
| Brilot AF, Lyon AS, Zelter A, Viswanath S, Maxwell A, MacCoss MJ, Muller EG, Sali A, Davis TN, Agard DA | 2021 | gTuRC(WT) open map | https://www.ebi.ac.uk/pdbe/entry/emdb/EMD-23636 | Electron Microscopy Data Bank, EMD-23636 |
| Brilot AF, Lyon AS, | 2021 | gTuSC dimer map | https://www.ebi.ac.uk/ | Electron Microscopy |

| Zelter A, Viswanath S, Maxwell A, MacCoss MJ, Muller EG, Sali A, Davis TN, Agard DA | | | | pdbe/entry/emdb/EMD-23639 | Data Bank, EMD-23639 |

The following previously published datasets were used:

| Author(s) | Year | Dataset title | Dataset URL | Database and Identifier |
|---|---|---|---|---|
| Zupa E, Pfeffer S | 2019 | Structure of the vertebrate gamma-Tubulin Ring Complex | https://www.rcsb.org/structure/6TF9 | RCSB Protein Data Bank, 10.2210/pdb6TF9/pdb |
| Howes SC, Geyer EA, LaFrance B, Zhang R, Kellogg EH, Westermann S, Rice LM, Nogales E | 2017 | Yeast tubulin polymerized with GTP in vitro | https://www.rcsb.org/structure/5W3F | RCSB Protein Data Bank, 10.2210/pdb5W3F/pdb |
| Aldaz HA, Rice LM, Stearns T, Agard DA | 2005 | Crystal Structure of gamma-tubulin bound to GTP | https://www.rcsb.org/structure/1Z5W | RCSB Protein Data Bank, 10.2210/pdb1Z5W/pdb |
| Zhang R, Alushin GM, Brown A, Nogales E | 2015 | Cryo-EM structure of GMPCPP-microtubule co-polymerized with EB3 | https://www.rcsb.org/structure/3JAL | RCSB Protein Data Bank, 10.2210/pdb3JAL/pdb |
| Wieczorek M, Urnavicius L, Ti S, Molloy KR, Chait BT, Kapoor TM | 2020 | Structure of gamma-tubulin in the native human gamma-tubulin ring complex | https://www.rcsb.org/structure/6V5V | RCSB Protein Data Bank, 10.1016/j.cell.2015.07.012 |
| Wieczorek M, Urnavicius L, Ti S, Molloy KR, Chait BT, Kapoor TM | 2020 | Structure of the native human gamma-tubulin ring complex | https://www.rcsb.org/structure/6V6S | RCSB Protein Data Bank, 10.2210/pdb6V6S/pdb |
| Wieczorek M, Urnavicius L, Ti S, Molloy KR, Chait BT, Kapoor TM | 2020 | Structure of GCP6 in the native human gamma-tubulin ring complex | https://www.rcsb.org/structure/6V6C | RCSB Protein Data Bank, 10.2210/pdb6V6C/pdb |
| Fong KK, Zelter A, Graczyk B, Hoyt JM, Riffle M, Johnson R, MacCoss MJ, Davis TN | 2018 | Novel phosphorylation states of the yeast spindle pole body. | http://www.yeastrc.org/fong_spb_2018/# | Yeast Resource Center Public Data Repository, fong_spb_2018 |
| Kollman JM, Greenberg CH, Li S, Moritz M, Zelter A, Fong KK, Fernandez JJ, Sali A, Kilmartin J, Davis TN, Agard DA | 2015 | Cryo-EM structure of gamma-TuSC oligomers in a closed conformation | https://www.emdataresource.org/EMD-2799 | EMDataResource, 2799 |
| Kollman JM, Greenberg CH, Li S, Moritz M, Zelter A, Fong KK, Fernandez JJ, Sali A, Kilmartin J, Davis TN, Agard DA | 2015 | Cryo-EM structure of gamma-TuSC oligomers in a closed conformation | https://www.rcsb.org/structure/5flz | RCSB Protein Data Bank, 10.2210/pdb5flz/pdb |

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

## Appendix 1

### Supplementary computational methods

#### Localizing Spc110 on γTuSC using integrative structure determination

The localization of Spc110 on γTuSC using integrative structure determination proceeded through four stages (*Figure 2—figure supplement 3*; *Alber et al., 2007*; *Rout and Sali, 2019*; *Russel et al., 2012*): (1) gathering data, (2) representing the system and translating data into spatial restraints, (3) structural sampling to produce an ensemble of structures that satisfies the restraints, and (4) analyzing and validating the ensemble structures and data. The modeling protocol (i.e., stages 2–4) was scripted using the *Python Modeling Interface* (PMI) package, a library for modeling macromolecular complexes based on our open-source *Integrative Modeling Platform* (IMP) package, version 2.8 (https://integrativemodeling.org;*Russel et al., 2012*). The current procedure is an updated version of previously described protocols (*Kim et al., 2018*; *Viswanath et al., 2017a*; *Wang et al., 2017*; *Webb et al., 2018*). Files containing the input data, scripts, and output results are available at https://salilab.org/gtuscSpc110 as well as the nascent Protein Data Bank archive for integrative structures (https://pdb-dev.wwpdb.org/) with deposition codes PDBDEV_00000077 (γTuSC monomer with Spc110p monomer), PDBDEV_00000078 (γTuSC monomer with Spc110p dimer), and PDBDEV_00000079 (γTuSC dimer with two Spc110p dimers).

#### Stage 1: gathering data

Chemical crosslinks between between Spc110 and γTuSC were identified by mass spectrometry of samples containing either Spc110$^{1-220}$-GCN4 dimer or Spc110$^{1-401}$-GST, informing the localization of Spc110 relative to γTuSC (*Figure 2—figure supplement 1*). γTuSC structure used was obtained from the PDB (code 5FLZ); it was determined primarily based on a cryo-EM density map of the disulfide-stabilized γTuSC filament at 6.9 Å resolution (EMDB code: 2799) (*Greenberg et al., 2016*; *Kollman et al., 2015*). Representation of Spc110$^{1-220}$ relied on (1) crystal structure of Spc110 NCC domain (*Figure 2B*) and (2) failure to detect related sequences of known structure in the rest of the Spc110 sequence by HHPred (*Söding et al., 2005*).

#### Stage 2: representing the system and translating data into spatial restraints

Information about the modeled system (above) can in general be used for defining its representation, defining the scoring function that guides sampling of alternative models, limiting sampling, filtering of good-scoring models obtained by sampling, and final validation of the models (*Figure 2—figure supplement 3*). Here, the 'flexible' representation for most of Spc110$^{1-220}$ reflects the absence of known related structures. The γTuSC and Spc110 NCC domain representations rely on their atomic structures. The scoring function relies on chemical crosslinks, excluded volume, and sequence connectivity.

An optimal representation facilitates accurate formulation of spatial restraints as well as efficient and complete sampling of good-scoring solutions, while retaining sufficient detail without overfitting, so that the resulting models are maximally useful for subsequent biological analysis (*Viswanath and Sali, 2019*). We first used a representation where a single γTuSC was bound to an Spc110$^{1-220}$ dimer. To maximize computational efficiency while avoiding using too coarse a representation, we represented the system in a multiscale fashion. A rigid body consisting of multiple beads was defined for γTuSC and the Spc110 NCC (Spc110$^{164-203}$). In a rigid body, the beads have their relative distances constrained during conformational sampling, whereas in a flexible string the beads are restrained by the scoring function (below). Rigid bodies were coarse-grained using one-residue beads, whose coordinates were those of the corresponding C$_\alpha$ atoms. The remaining regions in γTuSC without an atomic model were represented by a flexible string of beads encompassing 20 residues each. Due to lack of acceptable comparative models, and knowing that a large region of the N-terminus of Spc110 lacks secondary structure (*Figure 2A*), we used a flexible string of five-residue beads each to represent regions of Spc110$^{1-220}$ other than the coiled-coil domains. Additionally, we modeled a single γTuSC bound to an Spc110$^{1-220}$ monomer, as well as a complex of two adjacent γTuSCs each bound to an Spc110$^{1-220}$ dimer.

With this representation in hand, we next encoded the spatial restraints into a scoring function based on the information gathered in stage 1 as follows:

1. *Crosslink restraints*: The crosslinks (*Figure 2—figure supplement 1*) were used to construct the Bayesian scoring function (*Rieping et al., 2005*) that restrained the distances spanned by the crosslinked residues (*Shi et al., 2014*).
2. *Excluded volume restraints*: The excluded volume restraints were applied to each bead using the statistical relationship between the volume and the number of residues that it covered (*Alber et al., 2007*).
3. *Sequence connectivity restraints*: We applied the sequence connectivity restraints using a harmonic upper distance bound on the distance between consecutive beads in a subunit, with a threshold distance equal to twice the sum of the radii of the two connected beads. The bead radius was calculated from the excluded volume of the corresponding bead, assuming standard protein density (*Alber et al., 2007*; *Shi et al., 2014*).

## Stage 3: structural sampling to produce an ensemble of structures that satisfies the restraints

We aimed to maximize the precision at which the sampling of good-scoring solutions was exhaustive (stage 4). We sampled the positions of flexible Spc110 beads and the flexible linkers of γTuSC. The search for good-scoring models relied on Gibbs sampling, based on the Metropolis Monte Carlo algorithm (*Wang et al., 2017*). The positions of the γTuSC rigid body and the Spc110 NCC rigid body were fixed, while the initial positions of flexible γTuSC and Spc110 beads were randomized. The Monte Carlo moves included random translations of individual beads in the flexible segments of γTuSC and Spc110 (up to 3 Å). A model was saved every 10 Gibbs sampling steps, each consisting of a cycle of Monte Carlo steps that moved every moving bead once.

This sampling produced a total of 30 million models from 50 independent runs, requiring ~2 days on 200 CPU cores. For the most detailed specification of the sampling procedure, see the IMP modeling script (https://salilab.org/gtuscSpc110). We only consider for further analysis the ~3000,000 good-scoring models that satisfy the input datasets within their uncertainties (below).

## Stage 4: analyzing and validating the ensemble structures and data

Input information and output structures need to be analyzed to estimate structure precision and accuracy, detect inconsistent and missing information, and suggest more informative future experiments. We used the analysis and validation protocol published earlier (*Alber et al., 2007*; *Kim et al., 2018*; *Rout and Sali, 2019*; *Viswanath et al., 2017a*; *Viswanath et al., 2017b*): assessment began with the clustering of the models and estimating their precision based on the variability in the ensemble of good-scoring structures, and quantification of the structure fit to the input information. These validations are based on the nascent wwPDB effort on archival, validation, and dissemination of integrative structure models (*Burley et al., 2017*; *Sali et al., 2015*). We now discuss each one of these points in turn.

## 1. Clustering and structure precision

An ensemble of good-scoring structures needs to be analyzed in terms of the precision of its structural features (*Viswanath et al., 2017b*). The precision of a component position can be quantified by its variation in an ensemble of superposed good-scoring structures. It can also be visualized by the localization probability density for each of the components of the model.

As described above, integrative structure determination of the γTuSC-Spc110$^{1-220}$ dimer complex resulted in effectively a single good-scoring solution at the precision of 23.3 Å (*Figure 2—figure supplement 4*). The precision is the bead RMSD from the cluster centroid model averaged over all models in the cluster. Additionally, the sampling precisions for the γTuSC-Spc110$^{1-220}$ monomer complex and complex of two adjacent γTuSC-Spc110$^{1-220}$ dimers were 13.4 Å and 34.1 Å, respectively.

## 2. Fit to input information

An accurate structure needs to satisfy the input information used to compute it. The cluster of models of the γTuSC-Spc110$^{1\text{-}220}$ dimer complex satisfied 90.9% (92.8%) of the EDC (DSS) crosslinks; a crosslink is satisfied by a cluster of models if the corresponding Cα-Cα distance in any model in the cluster is less than 35 Å (25 Å) for DSS (EDC) crosslinks. Additionally, the γTuSC-Spc110$^{1\text{-}220}$ monomer complex satisfied 81.8% (80.9%) of EDC (DSS) crosslinks, and the complex of two adjacent γTuSC-Spc110$^{1\text{-}220}$ dimers satisfied 100% (94%) of EDC (DSS) crosslinks. The remainder of the restraints are harmonic, with a specified standard deviation. The cluster generally satisfied at least 95% of restraints of each type (excluded volume and sequence connectivity). A restraint is satisfied by a cluster of models if the restrained distance in any model in the cluster (considering restraint ambiguity) is violated by less than three standard deviations, specified for the restraint. Most of the violations are small and can be rationalized by local structural fluctuations, coarse-grained representation of the model, and/or finite structural sampling.

