## [Decision Letter]

**Acceptance summary:**

Microtubules are nucleated in yeast and metazoans by a γ-TuRC (γTuRC) complexes. This manuscript uses cryoEM and crosslinking mass-spec to show how the nucleating activator CM1 (centrosomin) motif in the yeast protein Spc110 binds to the yeast γTuRC. Models are proposed for how this leads to activation. A nice comparison is made with the metazoan γTuRC where activation appears to require additional components or proceed via a different mechanism.

**Decision letter after peer review:**

Thank you for submitting your article "CM1-driven assembly and activation of Yeast γ-Tubulin Small Complex underlies microtubule nucleation" for consideration by *eLife*. Your article has been reviewed by 2 peer reviewers, and the evaluation has been overseen by a Reviewing Editor and Anna Akhmanova as the Senior Editor. The reviewers have opted to remain anonymous.

The reviewers have discussed the reviews with one another and the Reviewing Editor has drafted this decision to help you prepare a revised submission.

Summary:

Microtubules are nucleated in yeast and metazoans by a γ-TuRC (γTuRC) complexes. This manuscript uses cryoEM and crosslinking mass-spec to show how the nucleating activator CM1 (centrosomin) motif, in the yeast protein Spc110, binds to the yeast γTuRC. Models are proposed for how this leads to activation. A nice comparison is made with the metazoan γTuRC where activation appears to require additional components or proceed via a different mechanism.

Essential revisions:

1. The introduction provides a good overview of previous structural studies of yeast γTuSC/γTuRC complexes, but does not quite introduce the background and open questions at the level of detail required to really appreciate the novelty of the Results. For example, a figure showing what is known from previous work about Spc110 binding to γTuSC/γTuRC could help to clarify background/question and novelty.

Currently one paragraph describes the structure of the metazoan γTuRC, but it is hard for the reader to gather how it differs from the yeast version. This paragraph could by simplified to remove some of the structural detail and more clearly highlight the differences with the yeast version. It would also be useful somewhere to discuss whether we would expect differences in their activation? I.e. compare the known roles of CM1 motif containing proteins.

2. The Results part is not an easy read for the non-expert. For example, some terms are not explained: EDC, DSS, Xrcc4, GCN4. The CM1 motif is not really introduced. The first figure begins by showing detailed structural information after a rather general introduction of the γTuRC complex, so the context of the shown structures with respect to the complex is not clear. The overview of crosslink data which introduces nicely the experiment is only in the Supplement which makes the figure less intuitive. Sometimes the description of the experiment could be clearer. For example, what does "reevaluating (…) filaments (…) with modern cryo-EM methods" mean? Are these new experiments or a new analysis of previous data. In Figure 2A and B, it is not clear why one structure is considered open and the other one closed; in fact the open structure looks more compact. Some paragraphs describe the structures in great detail, but do not conclude something, leaving the reader wonder what was learnt in terms of mechanism or conceptually, for example in "Conformational changes of Spc97p and Spc98 during assembly". The relative orientation of the shown structures is not always clear, sometimes it seems to change without indication, for example in Figure S16A,B.

3. The part describing potential phosphorylation sites was rather descriptive/speculative. For the sites with phenotypes in mutants, the mechanism how the phosphorylation (or absence of phosphorylation) induces a specific phenotype was unclear in most cases. The conclusion that most phosphorylations were predicted to weaken complex assembly seems to be in contrast to the Introduction where the impression was generated that phosphorylations promote function/activity.

4. The re-analysis of the recent human and *Xenopus* γTuRC structures was quite interesting as they implied a mechanism for human γTuRC activation by CDK5Rap2. However, the authors compared a *Xenopus* structure without γTuNA bound with a human structure with γTuNA bound. How can one conclude that the observed differences are indeed due to the presence vs absence of γTuNA and not due to species-specific differences? Does a comparison with another recently published human structure that did not contain γTuNA confirm the analysis of the authors?

5. Related to point 4 the manuscript needs more discussion of the possible mechanisms of activation in the metazoan γTuRCs. The authors describe the importance of a "closed" yeast γ-TuRC, mediated by CM1, to enable microtubule nucleation, whereas the Kapoor lab structure shows metazoan γ-TuRCs are "open" and asymmetric despite the presence of CM1. Moreover, recent work from the Petry lab suggests that nucleation of microtubules is initiated by a nucleus as little as 4 tubulin dimers, thus not requiring any "closed" γ-TuRC conformation. The current idea is therefore that γ-TuRC "closure" is a stochastic process, aided by lateral interactions between tubulin dimers that might "rectify" γ-TuRC conformation during the nucleation process. It would be helpful if the authors could also discuss this perspective.

---

## [Author Response]

Essential revisions:1. The introduction provides a good overview of previous structural studies of yeast γTuSC/γTuRC complexes, but does not quite introduce the background and open questions at the level of detail required to really appreciate the novelty of the Results. For example, a figure showing what is known from previous work about Spc110 binding to γTuSC/γTuRC could help to clarify background/question and novelty.Currently one paragraph describes the structure of the metazoan γTuRC, but it is hard for the reader to gather how it differs from the yeast version. This paragraph could by simplified to remove some of the structural detail and more clearly highlight the differences with the yeast version. It would also be useful somewhere to discuss whether we would expect differences in their activation? I.e. compare the known roles of CM1 motif containing proteins.

We have added an introductory figure to help introduce previously known information on Spc110p binding to γTuSCs. We have further revised the introduction as suggested by the reviewers, to clearly highlight differences between the metazoan TuRC vs the Yeast TuRC. However, we chose to keep a more detailed discussion on the role of CM1 proteins in the discussion, to maintain the focus on the Yeast TuRCs in the introduction.

2. The Results part is not an easy read for the non-expert. For example, some terms are not explained: EDC, DSS, Xrcc4, GCN4. The CM1 motif is not really introduced. The first figure begins by showing detailed structural information after a rather general introduction of the γTuRC complex, so the context of the shown structures with respect to the complex is not clear. The overview of crosslink data which introduces nicely the experiment is only in the Supplement which makes the figure less intuitive. Sometimes the description of the experiment could be clearer. For example, what does "reevaluating (…) filaments (…) with modern cryo-EM methods" mean? Are these new experiments or a new analysis of previous data. In Figure 2A and B, it is not clear why one structure is considered open and the other one closed; in fact the open structure looks more compact. Some paragraphs describe the structures in great detail, but do not conclude something, leaving the reader wonder what was learnt in terms of mechanism or conceptually, for example in "Conformational changes of Spc97p and Spc98 during assembly". The relative orientation of the shown structures is not always clear, sometimes it seems to change without indication, for example in Figure S16A,B.

We have added arrows to indicate the direction of rotation in figures. We have significantly revised the introduction to make main data figures easier to understand, including adding an introductory figure. We have slightly revised what was previously Figure 2 to make it clearer why one structure is closed vs open, and have included more detail on the cryo-EM experiments to make it clear that it is new data and a new processing scheme that led to improved resolution.

We agree with the reviewer that it is unfortunate that a thorough explanation of the cross-linking and modeling is only found in the Supplementary Methods. However, to simplify the text and make it more easily accessible, we made the decision to streamline the section on cross-linking to make it easier to understand by the non-expert. Many of the results presented in the cross-linking and modeling experiments are confirmed in greater detail by our structures, such that we decided to place a greater emphasis on the structures in the main text.

We agree that certain sections are primarily descriptive, especially when describing the conformational changes occurring between our structures. We believe it is important for the reader to have a detailed description of certain key aspects of our structures.

3. The part describing potential phosphorylation sites was rather descriptive/speculative. For the sites with phenotypes in mutants, the mechanism how the phosphorylation (or absence of phosphorylation) induces a specific phenotype was unclear in most cases. The conclusion that most phosphorylations were predicted to weaken complex assembly seems to be in contrast to the Introduction where the impression was generated that phosphorylations promote function/activity.

We agree that the section on phosphorylation sites can be described as rather speculative. In part, this is due to the fact that there are a lot of phosphorylation sites whose role is unclear, and by the fact that primarily phosphomimetic data is used to analyze the role of phosphorylation at each site.

Thus, we have added the following sentence:

“The role of many of these phosphorylation sites remains unclear as the phosphomimetic mutants used to investigate their function may not perfectly recapitulate the in vivo regulatory effects of the post-translational modifications.”

To avoid any misconception that phosphorylation must necessarily promote function, we now clearly state that all but one of the phosphorylation sites would be predicted to have an inhibitory role on assembly based on our structures. In places where we felt there may be some remaining ambiguity as to how phosphorylation might affect regulation, we have revised the text.

4. The re-analysis of the recent human and *Xenopus* γTuRC structures was quite interesting as they implied a mechanism for human γTuRC activation by CDK5Rap2. However, the authors compared a *Xenopus* structure without γTuNA bound with a human structure with γTuNA bound. How can one conclude that the observed differences are indeed due to the presence vs absence of γTuNA and not due to species-specific differences? Does a comparison with another recently published human structure that did not contain γTuNA confirm the analysis of the authors?

The recently published structure in Surrey et al. (2020) could have helped with our analysis in the manner described and indeed was carefully considered. However, atomic models for the structure are not publicly available, and the density between GCP2/6 is very poor, with the γ-tubulins being at low resolution. This unfortunately precludes a reliable analysis of the effect of CDK5RAP2 on the γ-tubulin conformation in humans, as suggested by the reviewer. We thus used the well conserved *Xenopus* structure as an example of a structure lacking CDK5RAP2.

5. Related to point 4 the manuscript needs more discussion of the possible mechanisms of activation in the metazoan γTuRCs. The authors describe the importance of a "closed" yeast γ-TuRC, mediated by CM1, to enable microtubule nucleation, whereas the Kapoor lab structure shows metazoan γ-TuRCs are "open" and asymmetric despite the presence of CM1. Moreover, recent work from the Petry lab suggests that nucleation of microtubules is initiated by a nucleus as little as 4 tubulin dimers, thus not requiring any "closed" γ-TuRC conformation. The current idea is therefore that γ-TuRC "closure" is a stochastic process, aided by lateral interactions between tubulin dimers that might "rectify" γ-TuRC conformation during the nucleation process. It would be helpful if the authors could also discuss this perspective.

We have revised the discussion to include some discussion of the results observed in recent work from the Petry and Surrey labs (Thawani, 2020; Consolati, 2020) that show either a 4-fold or a 7-fold cooperativity with ab-tubulin concentration. In our view, this indicates that as purified these are rather mediocre nucleators. While we do not specifically discuss the possibility that closure is a stochastic process potentially in which tubulin dimers play a role in rectifying the TuRC conformation, we do discuss the role that CM1 containing proteins could play in promoting nucleation.

The revised text is as follows:

“Recent work on microtubule nucleation from single purified metazoan γTuRCs has suggested microtubule nucleation remains a highly cooperative process, requiring ~4-7 αβ-tubulin dimers (Consolati et al., 2020; Thawani et al., 2020). […] Despite these major advances, significant gaps remain in our understanding of the how binding of regulatory proteins and PTMs act to modulate activation of yeast and metazoan γTuRCs.”